# Probabilistic Shapley Value Modeling and Inference

## Abstract

We propose probabilistic Shapley inference (PSI), a novel probabilistic framework to model and infer sufficient statistics of feature attributions in flexible predictive models, via latent random variables whose mean recovers Shapley values. PSI enables efficient, scalable inference over input-to-output attributions and their uncertainty, via a variational objective that jointly trains a predictive (regression or classification) model and its attribution distributions. To address the challenge of marginalizing over variable-length input feature subsets for Shapley value calculation, we introduce a masking-based neural network architecture, with a modular training and inference procedure. We evaluate PSI on synthetic and real-world datasets, showing that it achieves competitive predictive performance compared to strong baselines, while learning feature attribution distributions —centered at Shapley values— that reveal meaningful attribution uncertainty across data modalities.

## 1 Introduction

Learning feature attributions, i.e., quantifying how individual input features contribute to a (black-box) model's prediction, is a fundamental problem in interpretable and explainable machine learning (ML). Shapley values (Shapley et al., 1953), originating from cooperative game theory and the study of transferable utility, have become widely adopted in ML as a principled approach for feature attributions (Lundberg & Lee, 2017; Lundberg et al., 2020; Covert & Lee, 2021). For a supervised learning model, the Shapley value for feature $d$ is defined as its expected marginal contribution across all possible coalitions —where each coalition $S$ is a subset of features that excludes feature $d$— see Equation 3. Shapley value is the unique *removal based attribution method* (Covert et al., 2021) satisfying four key explainability axioms: Efficiency, Symmetry, Dummy (Null Player), and Additivity (Linearity); underpinning their appeal in model explanation (Merrick & Taly, 2020). We refer to Rozemberczki et al. (2022) for details on their theoretical foundation.

Despite its desirable properties and widespread application, most existing Shapley-based feature attribution methods operate on point estimates, thereby collapsing information in a dataset into a single summary statistic. This approach ignores the inherent uncertainty within a model's predictive distribution, which can be traced back to the observation's noise, overlapping classes, unobserved confounders, or feature correlations. For instance, when model predictions are uncertain, deterministic Shapley values may give incomplete information on the confidence in the importance of specific features. In addition, we are often not interested in identifying features with only high attributions, but also those with consistently low or negligible attributions (i.e., *dummy features*), so we can isolate (filter) and focus on the true drivers of a model's decisions.

Overlooking negligible or uncertain attribution information can lead to misleading or unstable attributions, of particular relevance in high-stakes domains such as healthcare or policymaking, where attribution reliability is paramount. Therefore, it is crucial to characterize (I) the inherent noise in the data, (II) the model's predictive uncertainty, and (III) how it all propagates to and impacts feature attributions.

In this work, we describe the attribution of a model's inputs to outputs through latent random variables, centered at Shapley values, via a probabilistic generative model of observed data. The model and the attribution distributions are jointly learnable through an efficient, stochastic variational inference procedure.

Our main contributions are:

1. **A Novel, Input-Conditional Prior Model for Shapley-Centered (Uncertain) Feature Attributions.** We introduce in Section 3 an input-conditional prior distribution over feature attributions, centered around Shapley values. We pose that the observed outputs are generated through a linear combination of attributions drawn from this Shapley prior that parameterize a Gaussian or a Bernoulli likelihood for regression and classification tasks, respectively (see Figure 1).

2. **Efficient Variational Inference for Simultaneous Model and Feature Attribution Learning.** We overcome challenges in model learning (non-analytical likelihoods and computationally demanding operations in feature size) by devising in Section 3.3 a variational inference framework for Probabilistic Shapley Inference (PSI). We derive a stochastic estimator of a (tighter) variational objective that enables efficient and scalable *joint learning* of the model and the Shapley prior.

3. **Efficient Processing of Variable Length Inputs.** A key challenge for removal-based attribution methods (e.g., Shapley values) lies in efficiently handling variable-length inputs and computing their marginals. In Section 3.4, we introduce a novel, masked embedding neural network (MENN) architecture that supports flexible modeling of variable-length, continuous and discrete inputs. MENN leverages the inherent parallelism of neural networks, enabling efficient computation on modern hardware (GPUs) by avoiding the need for sequential processing over feature subsets.

We evaluate and assess the *explainability and predictive performance* of the probabilistic Shapley inference (PSI) framework via empirical studies with the following aims:

1. **Objective 1.** To demonstrate that PSI recovers a distribution over attributions that closely reflects the true data-generating process, according to the Shapley value framework, via the empirical study presented in Section 4.1.

2. **Objective 2.** To showcase, through two illustrative case studies in Section 4.2, how probabilistic Shapley attributions that incorporate attribution uncertainty via coverage intervals provide more meaningful and informative explainability insights across multiple data modalities.

3. **Objective 3.** To establish, with results in Section 4.3, that the masking-based network presented in Section 3.4, which parameterizes the sufficient statistics of PSI's feature attribution distribution, produces more accurate predictive data estimates than standard baseline methods.

4. **Objective 4.** To verify PSI's competitive predictive performance across regression and classification tasks in Section 4.4, in par with several well-established explainable and black-box baselines.

## 2 Connections and Differences with Related Work

In this work, we address several core challenges in the application, computation and interpretation of feature attribution via Shapley values. We review and contextualize here the literature at the intersection of Shapley values and machine learning, with a focus on methods that address their computational, modeling and estimation challenges.

Despite its desirable properties and widespread application, Shapley value computation is burdened with inherent difficulties, due to its computational complexity (Van den Broeck et al., 2022), the difficulty in estimating model marginals (Covert et al., 2021), and their lack of uncertainty quantification.

Computation of exact Shapley values for a given feature $d$ is expensive, as it requires evaluating all possible feature coalitions, i.e., iterating over $S \in 2^{[D]\setminus\{d\}}$, where $2^{[D]\setminus\{d\}}$ represents the power set over $\{1, 2 \cdots, d-1, d+1, \cdots, D\}$. To mitigate this, recent work has framed the problem as a post-hoc weighted least squares estimation (Lundberg & Lee, 2017; Covert et al., 2020; Adebayo et al., 2021). While such methods offer useful insights into black-box model behavior, they come with notable limitations: they can be highly sensitive to small input perturbations (Ghorbani et al., 2019), may fail to faithfully reflect the decision boundaries of the original model (Rudin, 2019), and often require extensive sampling to produce reliable attributions (Jethani et al., 2021). FastSHAP (Jethani et al., 2021) addresses the computational bottleneck of Shapley estimation

by learning attributions in a single forward pass. However, it relies on training multiple neural networks in sequence, resulting in significant computational overhead. In contrast, we propose training *a single model* in which feature attributions are *embedded directly into the generative process*, rather than relying on post-hoc attributions.

A second challenge in Shapley value estimation is dealing with feature subsets of varying lengths. Typically, feature removal is treated as a marginalization process, where the model output is averaged over the omitted features. A common strategy is to marginalize removed features using an empirical or approximate input distribution. Existing methods approximate these marginals with distributions such as the product of independent input marginals, the empirical joint distribution, or uniform sampling over the input space (Datta et al., 2016; Lundberg & Lee, 2017; Strumbelj & Kononenko, 2010). However, these approximations can lead to misleading attributions when input features exhibit strong correlations. For a comprehensive discussion of marginalization strategies and their limitations, we refer the reader to Covert et al. (2021). Recent work has explored the use of baseline values as substitutes for marginalized features in Shapley value estimation: Sundararajan & Najmi (2020) proposed replacing missing features with fixed baselines, while Covert et al. (2021) showed that training models with independently replaced features can approximate marginalization under the input conditional distribution. Jethani et al. (2021) further extended this idea by selecting baseline values outside the observed data distribution to enhance distinguishability, i.e., to improve the model's ability to understand which values belong to the data distribution and which values are used as baselines. However, Covert et al. (2021) also highlighted the inherent difficulties of using baselines in continuous domains, where poorly chosen baselines can distort marginal estimates and lead to inaccurate attributions. In this work, we address these issues by *defining both feature embeddings and baseline values in a learned latent space*, and demonstrate that this leads to improved quality over prior approaches.

A third challenge and core limitation lies in the treatment of Shapley values as deterministic quantities. This view is influenced by the prevalence of non-probabilistic predictive ML models that produce single-point predictions —and explanations, typically related to a summary statistic of the data, such as the mean. However, this simplification can obscure important insights, underscoring the need to explicitly account for the uncertainty inherent in explanations (Chan et al., 2020; Alaa & Van Der Schaar, 2020; Lee et al., 2020). Although some post-hoc methods attempt to quantify uncertainty in Shapley estimates, they focus on capturing uncertainty through the variance estimation arising from ad-hoc approximations (Covert & Lee, 2021; Slack et al., 2021). Namely, these methods cannot propagate the predictive distribution's uncertainty to specific feature attributions; instead, they explain the uncertainty resulting from the Shapley value estimation procedure. Overall, they fail to reflect the uncertainty present in data, which we here characterize via a distribution over feature attributions. Recent work by Chau et al. (2023) is similar in spirit to ours, in that they assume a Shapley value centered Gaussian Process (GP) prior to, after observing some data, use the GP posterior to distribute its predictive uncertainty across feature attributions. However, our framework differs fundamentally: rather than providing post-hoc explanations under a GP prior, we put forward an end-to-end probabilistic learning framework that *jointly trains a predictive model and a conditional prior over feature attributions*.

## 3 Probabilistic Shapley Inference (PSI)

**Definitions and notations.** Let $\mathbb{X} \subseteq \mathcal{R}^D$ denote the input space of a machine learning model over $D$ features. We define a function $f_d$ that operates on variable-length subsets of features as

$$f_d : \bigcup_{S \subseteq [D]} \mathbb{X}_S \to \mathcal{R}, \quad \text{with} \quad f_d(\emptyset) = 0, \tag{1}$$

where $[D] = \{1, 2, \ldots, D\}$ is the index set of all input dimensions, $S \subseteq [D]$ is a feature subset, and $\mathbb{X}_S = \bigtimes_{d \in S} \mathbb{X}_d$ denotes the corresponding subspace. We define $\mathbb{X}_\emptyset$ as a singleton (e.g., $\{\emptyset\}$) to allow calls such as $f_d(\boldsymbol{x}_\emptyset) = f_d(\emptyset)$. Without loss of generality, we assume an additive output structure $f(\boldsymbol{x}_s) = \sum_{d=1}^{D} f_d(\boldsymbol{x}_s)$ over input subsets, and we write $f(\boldsymbol{x}) = \sum_{d=1}^{D} f_d(\boldsymbol{x})$ for the output given full input $\boldsymbol{x}$.

### 3.1 PSI's Generative Model: Conditional Shapley Priors

PSI characterizes feature attributions via probabilistic modeling, where we formulate the input to output attribution of each feature as a random variable conditioned on the input $\boldsymbol{x}$. Precisely, we assert a conditional prior distribution $\varphi_d | \boldsymbol{x}$ for the contribution of each feature $d$ of the input $\boldsymbol{x}$:

$$\varphi_d \mid \boldsymbol{x} \sim \mathrm{P}(\varphi_d \mid \boldsymbol{x}) = \mathcal{N}(\phi_d(\boldsymbol{x}), \sigma_d^2(\boldsymbol{x})) , \tag{2}$$

where the conditional mean $\phi_d(\boldsymbol{x})$ is defined by the Shapley-weighted marginal contribution of $f$, i.e.,

$$\phi_d(\boldsymbol{x}) = \sum_{S \subseteq [D] \setminus \{d\}} \mathrm{P}(S) \left( f(\boldsymbol{x}_{S \cup \{d\}}) - f(\boldsymbol{x}_s) \right), \quad \text{such that} \quad f(\boldsymbol{x}_s) = \int f(\boldsymbol{x}) \, \mathrm{dP}(\boldsymbol{x} \mid \boldsymbol{x}_s) , \tag{3}$$

with Shapley kernel $\mathrm{P}(S) = \frac{|s|!(D-|s|-1)!}{D!}$. That is, we ensure that the output of the generative model for a subset $\boldsymbol{x}_s$ aligns with the expected output under its corresponding marginalized input distribution.

We refer to $\mathrm{P}(\varphi_d \mid \boldsymbol{x})$ in Equation 2 as the **Shapley prior**, and denote samples $\varphi_d$ from this distribution as *stochastic Shapley values* (SSV). We write $\Phi = \{\varphi_d\}_{d=1}^D$ for the concatenation of SSVs for all input features.

By construction, $\varphi_d(\boldsymbol{x})$ is distributed around the Shapley values and satisfies the following properties.

**Theorem 3.1** (Axiomatic properties of $\phi(\boldsymbol{x})$). *The expected value $\phi(\boldsymbol{x})$ of the Shapley prior satisfies the following properties, by the definitions of $f(\boldsymbol{x}_s)$ and the Shapley kernel $P(S)$:*

**Property 1. (Efficiency)** $\sum_{d=1}^D \phi_d(\boldsymbol{x}) = f(\boldsymbol{x})$. *We start with* $\sum_{d=1}^D \sum_{S \subseteq [D] \setminus \{d\}} P(S) \left( f(\boldsymbol{x}_{S \cup \{d\}}) - f(\boldsymbol{x}_s) \right) = \sum_{T \subseteq [D]} c(T) f(\boldsymbol{x}_T)$, *where* $c(T) = \sum_{j \in T} P(T \setminus \{j\}) - \sum_{j \notin T} P(T)$, *which is 0 except for* $c(D) = 1$, *and* $C(\emptyset) = -1$. *Hence,* $\sum_{d=1}^D \phi_d(\boldsymbol{x}) = f(\boldsymbol{x}) - f(\emptyset) = f(\boldsymbol{x})$ *by definition in Equation 1.*

**Property 2. (Symmetry)** *if* $f(\boldsymbol{x}_{S \cup d_1}) = f(\boldsymbol{x}_{S \cup d_2})$, *for all* $s \in [D]$, *then* $\phi_{d_1}(\boldsymbol{x}) = \phi_{d_2}(\boldsymbol{x})$, *by application of Equation 3.*

**Property 3. (Dummy)** *if* $f(\boldsymbol{x}_{S \cup d}) = f(\boldsymbol{x}_s)$, *for all* $s \in [D]$, *then* $\phi_d(\boldsymbol{x})$ *is 0, by direct use of Equation 3.*

**Property 4. (Additivity)** *For two functions* $f^1$ *and* $f^2$, *the contribution to* $f = f^1 + f^2$ *of feature* $d$, *due to the linearity of the expectation, is* $\phi_d(\boldsymbol{x}) = \phi_d^1(\boldsymbol{x}) + \phi_d^2(\boldsymbol{x})$.

**Observation likelihood.** In practice, we do not observe random variables $\varphi_d$ —nor their realizations— but data (observations) that depend on these latent variables. In this work, we consider data likelihood functions for observations corresponding to regression and binary classification[1] tasks. We describe below and illustrate in Figure 1 the complete, Shapley prior-based probabilistic model by specifying the task-specific likelihoods that relate the latent function $f$ to observed data through $\varphi$.

For regression tasks with observations $y \in \mathcal{R}$, we adopt a Gaussian likelihood conditioned on latent variables $\varphi_d$:

$$y = \sum_{d=1}^D \varphi_d + \epsilon, \quad \epsilon \sim \mathcal{N}\left(\phi_0, \sigma_0^2\right) . \tag{4}$$

The marginal distribution of the observations, after integrating out the latent random variables over the Shapley prior, is analytically tractable:

$$y | \boldsymbol{x} \sim \mathcal{N}\left(\mu(\boldsymbol{x}), \sigma(\boldsymbol{x})\right), \quad \text{where} \quad \mu(x) = \phi_0 + f(\boldsymbol{x}) \quad \text{and} \quad \sigma^2(\boldsymbol{x}) = \sigma_0 + \sum_{d=1}^D \sigma_d^2(\boldsymbol{x}) , \tag{5}$$

by the **efficiency** property. Notice how the marginal distribution of the observations links $\phi_0 + f(\boldsymbol{x})$ to the expected value of the data $y$.

For classification tasks, we denote with binary indicator $\mathbf{1}_i$ whether observation $i$ is true (1) or false (0). We interpret $y$ as a latent logit random variable, with the observed label $\mathbf{1}_i$ defined by a Bernoulli likelihood parameterized by its logit $y_i$, i.e.,

$$\mathbf{1} \sim \mathrm{Bernoulli}\left(\mathrm{logits} = y\right) . \tag{6}$$

---

[1]Extensions to multi-class classification are left as future work.

Unfortunately, the classification's marginal likelihood is analytically intractable —we explain how we overcome this challenge in Section 3.3.

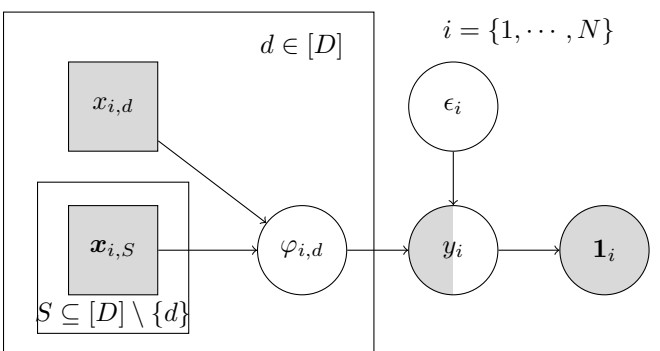

Figure 1: Graphical model (left) and data-generating procedure (DGP, on the right) for PSI's generative process. Shaded squares denote deterministic inputs, while shaded circles represent observed stochastic variables. Unshaded nodes correspond to latent variables. In regression, $y$ denotes the observed response; in classification, it represents latent logit values. Note that the first step of the DGP (on the right) involves a sum over subsets that is computationally, $\mathcal{O}\left(2^D\right)$, expensive.

For $i = \{1, \cdots, N\}$:

1. Calculate the expected value of per-feature $d \in [D]$ Shapley priors:

$$\phi_d(\boldsymbol{x}_i) = \sum_{S \subseteq [D] \setminus \{d\}} \mathrm{P}(S) \left( f(\boldsymbol{x}_{i, S \cup \{d\}}) - f(\boldsymbol{x}_{i,S}) \right)$$

2. Draw per feature $d \in [D]$ stochastic Shapley values, from their conditional Shapley priors:

$$\varphi_{i,d} | \boldsymbol{x}_i \sim \mathcal{N}\left( \phi_d(\boldsymbol{x}_i), \sigma_d^2(\boldsymbol{x}_i) \right)$$

3. Draw global data uncertainty terms:
$$\epsilon_i \sim \mathcal{N}\left( \phi_0, \sigma_0^2 \right)$$

4. Draw observed response (or latent logit) $y$:

$$y_i = \sum_{d=1}^{D} \varphi_{i,d} + \epsilon_i$$

(a) If classification, draw binary label:
$$\mathbf{1}_i \sim \mathrm{Bernoulli}\left( \text{logits} = y_i \right)$$

### 3.2 PSI Learning

In PSI's generative model, $f$ is an arbitrary functional connecting inputs to outputs. In practice, given $N$ input-output pairs, the goal is to identify an instance $f \in \mathcal{F}$ that best fits the observed data, where we denote the empirical dataset as $\mathbb{D} = (\boldsymbol{x}_i, y_i)_{i=1}^{N}$ for regression and use $\mathbb{D} = (\boldsymbol{x}_i, \mathbf{1}_i)_{i=1}^{N}$ for classification tasks.

To that end, one may either directly maximize the log-marginal likelihood of the dataset, or in a Bayesian fashion, compute the posterior over all the unknown latents, given the dataset. In any case, the marginal likelihood of observations is a critical function to compute.

Although the log-marginal over $\mathbb{D}$ for regression tasks

$$\mathcal{L}_{reg} = \log \prod_{i=1}^{N} \mathrm{P}(y_i \mid \boldsymbol{x}_i) = \log \prod_{i=1}^{N} \int_{\epsilon_i} \left( \int_{\varphi_{i,d}} \mathrm{P}(y_i \mid \Phi_i, \epsilon_i) \mathrm{P}(\epsilon_i) \, \mathrm{d}\epsilon_i \prod_{d=1}^{D} \mathrm{P}(\varphi_{i,d} \mid \boldsymbol{x}_i) \, \mathrm{d}\Phi_i \right) , \qquad (7)$$

is analytically tractable; the classification log-marginal likelihood

$$\mathcal{L}_{class} = \log \int_{y_i} \exp \left\{ \mathcal{L}_{reg} + \log \prod_{i=1}^{N} \mathrm{P}(\mathbf{1}_i \mid y_i) \right\} \mathrm{d}y_i , \qquad (8)$$

is analytically intractable.

More importantly, the computation of the attributional distribution $\varphi_d \mid \boldsymbol{x}$, which we would like to use for model interpretability, involves **evaluating an exponential number of feature subsets**, with a computational cost of $\mathcal{O}\left(2^D\right)$ —a well-known bottleneck in Shapley value estimation that hinders direct calculation of the marginal likelihoods in Equations 7 and 8.

To overcome these issues, we adopt a variational framework that enables joint model learning and tractable inference of the stochastic Shapley values.

Precisely, we aim to *simultaneously learn* the unknown functional $f$, along with the global bias term $\phi_0$ and the uncertainty sources in the data: the input conditional (heteroscedastic) uncertainty $\sigma(\boldsymbol{x})$ and the observation's homoscedastic uncertainty $\sigma_0$. Furthermore, and leveraging the probabilistic nature of the model, we are also interested in inference of the latent random variables $\varphi_d \mid \boldsymbol{x}$, *to facilitate interpretable attributions of each input instance, efficiently.* We denote the full set of learnable model parameters with $\theta$, which includes parameters of functionals $f(\boldsymbol{x})$ and $\sigma(\boldsymbol{x})$ (e.g., neural network weights and biases), the global bias term $\phi_0$, and the homoscedastic variance term $\sigma_0$.

### 3.3 Variational PSI

We hereby introduce a variational approximation to the otherwise intractable posterior over latent variables, by computing the evidence lower-bound (ELBO) for regression and classification tasks:

$$\mathcal{L}_{reg} \geq \mathcal{L}_{reg}^{ELBO} = \sum_{i=1}^{N} \mathbb{E}_{Q_{shap_i}} \left\{ \log \frac{P(y_i, \Phi_i \mid \boldsymbol{x}_i)}{Q_{shap_i}} \right\} , \tag{9}$$

$$\mathcal{L}_{class} \geq \mathcal{L}_{class}^{ELBO} = \sum_{i=1}^{N} \mathbb{E}_{Q_{logit_i} Q_{shap_i}} \left\{ \log \frac{P(\mathbf{1}_i, y_i, \Phi_i \mid \boldsymbol{x}_i)}{Q_{logit_i} Q_{shap_i}} \right\} , \tag{10}$$

where we define variational families

$$Q_{shap_i} = Q_{shap_i}(\Phi_i \mid \boldsymbol{x}_i) = \prod_{d=1}^{D} Q_{shap_d}(\varphi_{i,d} \mid \boldsymbol{x}_i) \quad \text{with} \quad Q_{shap_d}(\varphi_d \mid \boldsymbol{x}) = \mathcal{N}(\tilde{f}_d(\boldsymbol{x}), \tilde{\sigma}_d^2(\boldsymbol{x})) , \quad \text{and} \tag{11}$$

$$Q_{logit_i} = Q_{logit_i}(y_i \mid \boldsymbol{x}_i, \mathbf{1}_i) = P(y_i \mid y_i > 0, \boldsymbol{x}_i)^{\mathbf{1}_i} P(y_i \mid y_i \leq 0, \boldsymbol{x}_i)^{1-\mathbf{1}_i} . \tag{12}$$

The variational Shapley distribution $Q_{shap_i}$ —the critical component of the probabilistic Shapley inference (PSI) framework— is shared between the regression and classification tasks.

For classification, we incorporate additional probabilistic components that model the latent logit random variable (the logarithm of the odds) of the binary observation probability. Precisely, we condition the logit's distribution on their sign, as determined by the observed binary variable $\mathbf{1}$: logits are greater than zero if $\mathbf{1} = 1$, and less than or equal to zero otherwise (i.e., $\mathbf{1} = 0$). This extra element of the classification variational posterior, which allows for information flow from the binary indicator random variables to posterior over logits, is a truncated normal distribution over $y$, i.e.,

$$P(y \mid y \geq 0, \boldsymbol{x})^{\mathbf{1}} P(y \mid y \leq 0, \boldsymbol{x})^{1-\mathbf{1}} = \frac{\mathcal{N}\left(\mu(\boldsymbol{x}), \sigma^2(\boldsymbol{x})\right)}{\Psi(\mathbf{1}, \boldsymbol{x})} , \tag{13}$$

$$\text{with} \quad \Psi(\mathbf{1}, \boldsymbol{x}) = \frac{1}{2} \left( 1 + \mathrm{erf}\left\{ \frac{\mu(\boldsymbol{x})}{\sigma(\boldsymbol{x})} \right\} \right)^{\mathbf{1}} \left( 1 + \mathrm{erf}\left\{ -\frac{\mu(\boldsymbol{x})}{\sigma(\boldsymbol{x})} \right\} \right)^{1-\mathbf{1}} , \tag{14}$$

given logit sufficient statistics $\mu(\boldsymbol{x})$ and $\sigma(\boldsymbol{x})$ as in Equation 5, after marginalization of the Shapley prior.

Before delving into additional details and technicalities of PSI's variational inference procedure, we rewrite the regression ELBO in terms of the variational Shapley distribution $Q_{shap_i}$ as

$$\mathcal{L}_{reg} \geq \sum_{i=1}^{N} \underbrace{\mathbb{E}_{Q_{shap_i}} \left\{ \log P(y_i \mid \Phi_i) \right\}}_{(i)} - \underbrace{D_{\mathrm{KL}}\left( Q_{shap_i} \| P(\Phi_i \mid \boldsymbol{x}_i) \right)}_{(ii)} , \tag{15}$$

highlighting that $(i)$ the first term fits the observed data, while $(ii)$ the second term finds a variational distribution $Q_{shap_i}$ that approximates the Shapley prior $P(\Phi_i \mid \boldsymbol{x}_i)$. Following standard variatonal inference, one would optimize the ELBOs in Equations 9 and 10 with respect to variational parameters in Equations 11 and 12, which we collectively denote with $\lambda$. In contrast, we propose a revision to the lower-bounds of PSI, based on a careful choice of the conditional variational families $Q_{shap_d}(\varphi_d \mid \boldsymbol{x})$.

### 3.3.1 A Tighter and Computationally Efficient ELBO for PSI

To devise a tighter, and computationally efficient computation of PSI's variational lower-bound, we pose the same functional form for the variational and prior functions, i.e., $\tilde{f}_d(\boldsymbol{x}) = f_d(\boldsymbol{x})$, and uncertainty functionals $\tilde{\sigma}_d(\boldsymbol{x}) = \sigma_d(\boldsymbol{x})$. Recall that we are not equating the conditional Shapley prior $\mathrm{P}(\phi_d \mid \boldsymbol{x}) = \mathcal{N}\left(\phi_d(\boldsymbol{x}), \sigma_d^2(\boldsymbol{x})\right)$ to the variational family $\mathrm{Q}_{shap_d}(\varphi_d \mid \boldsymbol{x}) = \mathcal{N}(f_d(\boldsymbol{x}), \sigma_d^2(\boldsymbol{x}))$, but imposing a shared functional parameterization between the generative model and the variational family. This shared functional parameterization enables us to establish the following proposition.

**Proposition 3.2.** *For $Q_{shap_d}(\varphi_d \mid \boldsymbol{x}) = \mathcal{N}(f_d(\boldsymbol{x}), \sigma_d^2(\boldsymbol{x}))$ and any function $h$, the **efficiency** property of Shapley values guarantees that $\mathbb{E}_{Q_{shap_i}}\left\{h\left(\phi_0 + \sum_{d=1}^{D} \varphi_{i,d}\right)\right\}$ is equal to $\mathbb{E}_{P(\Phi_i \mid \boldsymbol{x}_i)}\left\{h\left(\phi_0 + \sum_{d=1}^{D} \varphi_{i,d}\right)\right\}$.*

*Proof.* The proof is followed by the efficiency property and is detailed in Appendix A. □

Equipped with this proposition, we can now equate expectations over the variational Shapley distribution $\mathrm{Q}_{shap_i}$ with those over the true prior, to compute the first term of the ELBO in Equation 15 as $\mathbb{E}_{\mathrm{Q}_{shap_i}}\{\mathrm{P}(y_i \mid \Phi_i)\} = \mathbb{E}_{\mathrm{P}(\Phi_i \mid \boldsymbol{x}_i)}\{\mathrm{P}(y_i \mid \Phi_i)\} = \int_{\Phi_i} \mathrm{P}(y_i \mid \Phi_i)\mathrm{P}(\Phi_i \mid \boldsymbol{x}_i)\,\mathrm{d}\Phi_i = \mathrm{P}(y_i \mid \boldsymbol{x}_i)$.

**Tighter variational PSI ELBO.** We now formulate a modified ELBO for PSI:

$$\mathcal{L}_{reg}^{ELBO} \leq \mathcal{V}_{reg} = \sum_{i=1}^{N} \log \mathrm{P}(y_i \mid \boldsymbol{x}_i) - \beta D_{\mathrm{KL}}\left(\mathrm{Q}_{shap_i}\|\mathrm{P}(\Phi_i \mid \boldsymbol{x}_i)\right) \leq \mathcal{L}_{reg}\,, \qquad (16)$$

with hyperparameter $0 \leq \beta \leq 1$, which regulates the tightness of the lower-bound ($\mathcal{V}_{reg} = \mathcal{L}_{reg}$, for $\beta = 0$) by controlling the influence of the second term in the loss. Additionally, we use this new lower-bound, $\mathcal{V}_{reg}$, to rewrite the ELBO for the binary classification task (see detailed derivations in Appendix B):

$$\mathcal{L}_{class}^{ELBO} \leq \mathcal{V}_{class} = \sum_{i=1}^{N} \underbrace{\mathbb{E}_{\mathrm{Q}_{logit_i}}\{\log \mathrm{P}(\mathbf{1}_i \mid y_i) + \log \mathrm{P}(y_i \mid \boldsymbol{x}_i)\}}_{(\mathrm{I})} + \underbrace{\mathcal{H}(\mathrm{Q}_{logit_i})}_{(\mathrm{II})} - \underbrace{\beta D_{\mathrm{KL}}\left(\mathrm{Q}_{shap_i}\|\mathrm{P}(\Phi_i \mid \boldsymbol{x}_i)\right)}_{(\mathrm{III})}$$

$$\qquad (17)$$

$$\leq \mathcal{L}_{class}\,, \qquad (18)$$

where $\mathcal{H}(\mathrm{Q}_{logit_i})$ is the entropy of the variational logit distribution. This PSI classification objective promotes (I) learning input-conditional predictive logits $y_i$, marginalized over SSVs, that explain the observed binary labels $\mathbf{1}_i$; (II) high-entropy variational distributions over logits, to avoid overconfident or degenerate solutions; and (III) fitting a tractable variational $\mathrm{Q}_{shap_i}$ distribution over SSVs.

**The subset marginal constraint for Shapley computations.** Maximizing $\mathcal{V}$ (for regression or classification) encourages learning parametric functionals $f_d$ that, due to PSI model design, directly satisfy Property 1. To meet the remaining properties in Theorem 3.1, it is mandatory to fulfill the subset-marginal constraint $f_d(\boldsymbol{x}_s) = \int f_d(\boldsymbol{x})\,\mathrm{dP}(\boldsymbol{x} \mid \boldsymbol{x}_s)$, where under slight abuse of notation, we denote with $\mathrm{P}(\boldsymbol{x} \mid \boldsymbol{x}_s)$ the distribution of features not in subset $S$. Namely, we must enforce that the model's output for a subset $\boldsymbol{x}_s$ aligns with the model's expected output under the corresponding marginalized input distribution. In training, we enforce this constraint by randomly masking the features in $\boldsymbol{x}$, in a similar fashion to Jethani et al. (2021), where the feature subsets were replaced with values outside the support of the data distribution. In Appendix C, we show that under reasonable assumptions, random feature removal during the variational training procedure with $\mathcal{V}_{reg}$ and $\mathcal{V}_{class}$ guarantees that this constraint is satisfied.

**Efficient computation of $\mathcal{V}_{reg}$ and $\mathcal{V}_{class}$.** We describe below how the tighter ELBOs of Equations 16 and 18 can be computed through closed-form expressions that enable their scalable optimization.

- **The Shapley Kullback-Leibler divergence.** The key term $(ii)$ in Equation 15 obeys

$$D_{SHAP} = D_{\mathrm{KL}}\left(\mathrm{Q}_{shap_d}(\Phi \mid \boldsymbol{x}) \| \mathrm{P}(\Phi \mid \boldsymbol{x})\right) = \sum_{d=1}^{D} \frac{\left(\left[\sum_{S \subseteq [D]\setminus\{d\}} \mathrm{P}(S)\left(f(\boldsymbol{x}_{S\cup d}) - f(\boldsymbol{x}_s)\right)\right] - f_d(\boldsymbol{x})\right)^2}{2\sigma_d(\boldsymbol{x})^2}.$$ 
(19)

Instead of its exact computation (of exponential complexity due to the sum over coalitions), we derive a computationally efficient, *stochastic estimate* as follows:

$$\hat{D}_{SHAP} = D \frac{\left|\left(\frac{\sum_{k=1}^{K_1} f(\boldsymbol{x}_{s_{1,k}\cup d}) - f(\boldsymbol{x}_{s_{1,k}})}{K_1} - f_d(\boldsymbol{x})\right)\left(\frac{\sum_{k'=1}^{K_2} f(\boldsymbol{x}_{s_{2,k'}\cup d}) - f(\boldsymbol{x}_{s_{2,k'}})}{K_2} - f_d(\boldsymbol{x})\right)\right|}{2\sigma_d(\boldsymbol{x})^2},$$ 
(20)

with $s_{1,k}$ , $s_{2,k'} \sim \mathrm{P}(S)$ and $d \sim \mathrm{U}(1, D)$. The theoretical properties of this estimator $\hat{D}_{SHAP}$ are detailed in Appendix D. In practice, we use a single sample for each subset —$s_{1,k}$ and $s_{2,k}$ (i.e., $K_1 = K_2 = 1$) —along with a single sample over the feature dimension $d$ to ensure computational efficiency.

**Remark 3.3.** $D_{SHAP} \to 0$ *implies $f_d \approx \phi_d$. A property of $\phi_d$ is that $\mathbb{E}_{\boldsymbol{x}\sim\mathbb{D}}\{\phi_d(\boldsymbol{x})\} \approx \mathbb{E}_{\boldsymbol{x}\sim\mathbb{D}}\{f_d(\boldsymbol{x})\} \approx 0$. This serves as a useful diagnostic tool during training: the data-average of the variationally trained functional's output $f_d$ should converge toward zero per training iteration. We illustrate this behavior in Figure 8 of Appendix G.*

- **Observation likelihood terms.** We leverage standard gradient reparameterizations for the expectation of the regression likelihood, and the inverse transform sampling (Bou et al., 2023) for the unbiased estimation of the expectation of Bernoulli log-likelihoods $\mathbb{E}_{\mathrm{Q}_{logit}}\{\log\mathrm{P}(\boldsymbol{1}\mid y)\}$ —term (I) in Equation 18.

- **Classification entropy and cross-entropy terms.** The classification entropy term $\mathcal{H}(\mathrm{Q}_{logit}) = \mathbb{E}_{\mathrm{P}(y|y>0,\boldsymbol{x})^1 \mathrm{P}(y|y\le 0,\boldsymbol{x})^{1-1}}\left\{-\log\mathrm{P}(y\mid y>0,\boldsymbol{x})^{\boldsymbol{1}}\mathrm{P}(y\mid y\le 0,\boldsymbol{x})^{1-\boldsymbol{1}}\right\}$ corresponds to the entropy of a truncated normal distribution, with known closed-form solution, given logit sufficient statistics $\mu(\boldsymbol{x})$ and $\sigma(\boldsymbol{x})$ as in Equation 5:

$$\mathcal{H}(\mathrm{Q}_{logit}) = \frac{1}{2}\left(\log\left\{2\pi e\sigma^2(\boldsymbol{x})\Psi(\boldsymbol{1},\boldsymbol{x})\right\} + \Omega(\boldsymbol{1},\boldsymbol{x})\right), \quad \text{where} \quad \Omega(\boldsymbol{1},\boldsymbol{x}) = (-1)^{\boldsymbol{1}}\cdot\frac{\frac{\mu(\boldsymbol{x})}{\sigma(\boldsymbol{x})}\frac{1}{\sqrt{2\pi}}\exp\left\{-\frac{\mu^2(\boldsymbol{x})}{\sigma^2(\boldsymbol{x})}\right\}}{\Psi(\boldsymbol{1},\boldsymbol{x})}.$$ 
(21)

Additionally, we compute the cross-entropy term in Equation 18 as

$$\mathbb{E}_{\mathrm{Q}_{logit}}\{\log\mathrm{P}(y\mid\boldsymbol{x})\} = \frac{1}{2}\log 2\pi\sigma^2(\boldsymbol{x}) + \frac{1}{2\sigma^2(\boldsymbol{x})}\left(v^2(\boldsymbol{1},\boldsymbol{x}) + (m(\boldsymbol{1},\boldsymbol{x}) - \mu(\boldsymbol{x}))^2\right),$$ 
(22)

where $m$ and $v^2$ are the mean and variance statistics of the truncated normal distribution given by:

$$m(\boldsymbol{1},\boldsymbol{x}) = \mu(\boldsymbol{x}) - \frac{\sigma^2(\boldsymbol{x})}{\mu(\boldsymbol{x})}\Omega(\boldsymbol{1},\boldsymbol{x}) \quad \text{and} \quad v^2(\boldsymbol{1},\boldsymbol{x}) = \sigma^2(\boldsymbol{x})\left(1 - \Omega(\boldsymbol{1},\boldsymbol{x}) - \left(\frac{\sigma(\boldsymbol{x})}{\mu(\boldsymbol{x})}\Omega(\boldsymbol{1},\boldsymbol{x})\right)^2\right).$$ 
(23)

**Efficient and scalable PSI learning procedure.** We summarize PSI's learning procedure, according to the ELBOs of Equations 16 and 18, in Algorithm 1.

## 3.4 Neural Network-based Shapley Prior Sufficient Statistics: The $f_d$ and $\sigma_d$ networks

In this section, we describe how we leverage *neural networks to model the sufficient statistics of the Shapley prior* in Equations 1 and 2, i.e., $f_d$ and $\sigma_d$. In Section 3.4.1, we also describe the proposed masked embedding neural network (MENN) architecture that enables efficient *variable length input* (VLI) modeling.

---

**Algorithm 1** SGD-based variational PSI learning using a mini-batch size of $M \leq N$.

---

1: **Input:** Dataset $\mathcal{D}$, model parameters $\theta$, batch size $M$, and feature removal probability $p$
2: **while** not converged **do**
3:     *1. **Sample features** for marginal integral constraint (described by the binary matrix $\mathbf{R}$ in Figure 2)*
4:       Draw a subset of feature indices for each instance $i$ as:
5:           $\mathcal{S}_i = \{d | \mathbf{1}_d = 1, \forall d \in [D]\}$, where $\mathbf{1}_d \sim \text{Bernoulli}(p)$
6:     *2. **Sample** variable length **observed data** using $\mathcal{S}_i$*
7:       Draw $\{(\mathbf{x}_{\mathcal{S}_i, i}, y_i)\}_{i=1}^M \sim \mathcal{D}$
8:     *3. **Compute stochastic estimate** of $\hat{\phi}$*
9:       *3.a Sample per instance $s_1$ and $s_2$ coalitions (given $\mathcal{S}_i$)*
10:        Draw $\{s_{i,1,k_1}, s_{i,2,k_2}\}_{i=1}^M \sim \frac{|s|!(|\mathcal{S}_i| - |s| - 1)!}{|\mathcal{S}_i|!}$ for each data instance $i$       ▷ Note that we use $K_1 = K_2 = 1$
11:      *3.b Sample feature indices $d \in \mathcal{S}_i$*
12:        Draw $\{d_i\}_{i=1}^M \sim \frac{1}{|\mathcal{S}_i|}$ uniformly for each instance $i$.
13:      *3.c Compute $\hat{\phi}$ as in Equation 20*       ▷ Use $s_{i,1,k_1}, s_{i,2,k_2}$, and $d_i$ for per-instance $i$.
14:    *4. **Compute PSI loss ($\mathcal{V}$) and its gradients** using the closed form solutions*
15:      $g \leftarrow \nabla_\theta \mathcal{V}$
16:    *5. **Update PSI model parameters***
17:      $\theta \leftarrow$ Update using gradients $g$
18: **end while**
19: **Output:** PSI model, with parameters $\theta$

---

We start by clarifying why our framework is based on embedded baselines for marginal estimation. Recent methods estimate marginal contributions by modifying the input space with baseline values. For example, Yoon et al. (2018) simulate feature removal by replacing input features with a fixed zero vector.

However, using constant baselines in the input space can lead to misleading attributions —especially in high-dimensional or structured data. A more principled alternative, proposed by Jethani et al. (2021), is to explicitly append a feature removal mask $\boldsymbol{R}$, allowing a neural network to distinguish between original and substituted inputs.

We follow a similar technique by operating in a (learned) embedded space. Rather than operating directly in the input space, we map each feature $x_d$ to a high-dimensional embedding $\boldsymbol{z}_d$, and substitute missing features with their baseline embeddings $\boldsymbol{b}_d$.

We begin by mapping the inputs $\boldsymbol{x} \in \mathbb{X}$ to $\mathbf{Z} \in \mathcal{R}^{D \times D_z}$, where each row of $\mathbf{Z}$ represents $\boldsymbol{z}_d = MENN_1(x_d) \in \mathcal{R}^{D_z}$. Because sequentially iterating over all $d \in [D]$ is computationally inefficient, we employ a masked network architecture that performs these operations in parallel, as described in Section 3.4.1. When marginalizing over $x_d$, i.e., using variable length input that does not include feature $x_d$ described by *feature removal matrix $\boldsymbol{R}$*, we replace the embedding $\boldsymbol{z}_d$ of $x_d$ with the learnable baseline vector $\mathbf{b}_d$ to simulate feature removal, and construct $\boldsymbol{z}_d'$. We then take a linear combination of these embeddings using an affine transformation matrix $\mathbf{W}$ and reduce the dimensionality from $D \times D_z$ to $D_z$. We input the result into a feedforward neural network (FFNN) to obtain $f_d(\boldsymbol{x})$.

For $\sigma_d(\boldsymbol{x})$, we want to associate each per input feature dimension $x_d$ and its corresponding output $f_d(x_d)$ with their uncertainty. In order to leverage the learned (variable input) embeddings and associate them with their corresponding functional output $f_d$, we concatenate $\boldsymbol{z}_d'$ with $\boldsymbol{o}_d = MENN_2(f_d)$, and use a feedforward neural network (FFNN) to obtain $\sigma_d(\boldsymbol{x})$. To avoid this uncertainty module altering the learned feature embeddings and their outputs, we freeze $\boldsymbol{z}_d'$ and $f_d$ when inputting them to the FNNN that outputs $\sigma_d$, following a similar approach as in (Stirn et al., 2023) and more recently (Bramlage & Curio, 2025). We showcase the overall neural network architecture to compute Shapley prior sufficient statistics in Figure 2.

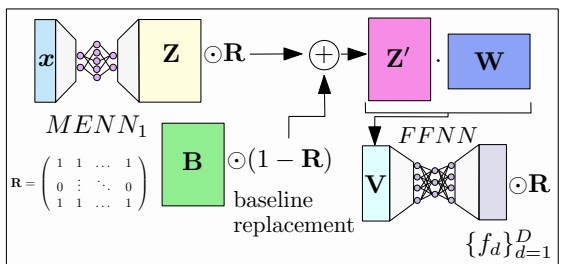 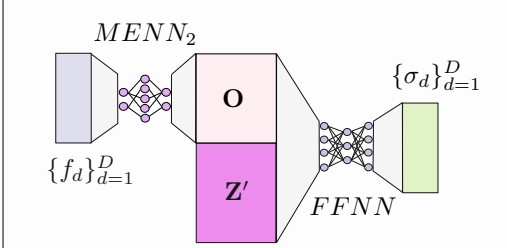

Figure 2: The $f_d$ and $\sigma_d$ networks. We independently generate latent embedding vectors for each feature, and replace absent features with a baseline vector in the embedding space. We compute a linear combination of these embeddings to arrive at a single vector, which we use as model output $f_d$. To compute the variance $\sigma_d$, we re-utilize the information in the model predictions along with the feature representations. Note that, $f(\emptyset) = 0$ can easily be satisfied by multiplying the final output by $\boldsymbol{R}$.

### 3.4.1 The Masked Embedding Neural Network (MENN) Architecture

We present here the masked-neural architecture used for efficient computation of (subset marginal) sufficient statistics. While masked neural architectures have been widely used across domains such as natural language processing and density estimation (Vaswani et al., 2017; Devlin et al., 2018; Papamakarios et al., 2017), our goal here is distinct: i.e., the efficient computation of high-dimensional per-feature embeddings in parallel —avoiding the need to iterate over individual feature dimensions. The masked embedding neural network (MENN) architecture we describe here enables efficient information flow across neurons while generating continuous per-feature embeddings in parallel —see an illustrative example in Appendix E.

A $K$-layer neural network can be represented by $\boldsymbol{z}_k = act_k(\boldsymbol{z}_{k-1}\mathbf{W}_k)$, where $\boldsymbol{z}_0 = \boldsymbol{x}^\top$ and $\boldsymbol{z}_L = \hat{y}$. We define each $\mathbf{W}_k = \mathbf{W}'_k \odot \mathbf{M}_k$, where $\mathbf{M}_k$ are binary $D_1^k \times D_2^k$ masking matrices with $D_2^k \geq D$. The permeability of the network is determined by the non-zero elements of $\mathbf{M}_{1:K} = \prod_{k=1}^K \mathbf{M}_k$. In particular, if $\mathbf{M}_{1:K}(d,l) > 0$, then there is an information flow from $d^{\text{th}}$ feature to $l^{\text{th}}$ output neuron (Germain et al., 2015).

We begin by defining the MENN initial masking matrix $\mathbf{M}_1$ as

$$\mathbf{M}_1(d,l) = \begin{cases} 1 & \text{if } 1 + (d-1)e_1 \leq l \leq de_1\mathbf{1}_{d \neq D} + D_2^1\mathbf{1}_{d=D} \\ 0 & \text{otherwise,} \end{cases}$$

and $\mathbf{M}_k$, for $k > 1$, as $col_l(\mathbf{M}_k) = sgn(row_d(\mathbf{M}_{1:k-1})^\top)$, for $1 + (d-1)e_k \leq j \leq de_k\mathbf{1}_{d \neq D} + D_2^k\mathbf{1}_{d=D}$, where $sgn(\cdot)$ is the sign function, $e_k = nint(D_2^k/D)$ for $d \in [D]$, $k \in [K]$, and $D_2^K$ is an integer multiple of $D$. Here, $nint(.)$ refers to the nearest integer function. The multiplication of a sequence of such matrices has the following recursion:

$$col_l(\mathbf{M}_{1:k}) = \gamma_d col_{1+(d-1)e_{k-1}}(\mathbf{M}_{1:k-1}), \tag{24}$$

where $1 + (d-1)e_k \leq l \leq de_k\mathbf{1}_{d \neq D} + D_2^k\mathbf{1}_{d=D}$, and $\gamma_d \in \mathbb{N}^+$ for $d \in [D]$. Since $D_2^K$ is an integer multiple of $D$, the columns of $\mathbf{M}_{1:K}$ repeat $col_{1+(d-1)e_{K-1}}(\mathbf{M}_{1:K-1})$ $e_K$ number of times. Then, $D \times De_K$ matrix $M_{1:K}$ obeys:

$$\mathbf{M}_{1:K}(d,l) = \begin{cases} \gamma_d, \text{ for } \gamma_d > 0, & \text{if } 1 + (d-1)e_K \leq l \leq de_K \\ 0, & \text{otherwise.} \end{cases}$$

For instance, for $D = 3, D_2^K = 6$ (i.e., $D_z = e_K = 2$), $\mathbf{M}_{1:K}$ is (see a step-by-step example in Appendix E):

$$\prod_{k=1}^K M_k = \begin{bmatrix} \gamma_1 & \gamma_1 & 0 & 0 & 0 & 0 \\ 0 & 0 & \gamma_2 & \gamma_2 & 0 & 0 \\ 0 & 0 & 0 & 0 & \gamma_3 & \gamma_3 \end{bmatrix}.$$

We emphasize that $\mathbf{M}_{1:K}$ in MENN is a masking operation applied to neural network weights, to efficiently map each input feature to a different embedding. On the contrary, $\boldsymbol{R}$ as depicted in Figure 2 is a feature removal matrix for baseline embedding replacement, i.e., it discards the MENN generated embeddings if a feature is absent and modulates their replacement with a baseline vector.

# 4 Results

## 4.1 PSI Faithfully Captures the Data Generating Process' Feature Attribution Distribution

To illustrate PSI's capabilities in capturing true feature attributions, we generate synthetic datasets via the following process: (I) draw independent features $x_1, x_2, x_3 \sim \mathrm{U}(-4, 4)$; (II) sample feature-conditional SSVs $\varphi_1 \sim \mathcal{N}\left(\exp\{-x_1^2\} - \frac{\sqrt{\pi}}{8}\mathrm{erf}\{4\}, 0.6\cos{(0.03x_1)}^{800}\right)$, $\varphi_2 \sim \mathcal{N}\left(\sin\{-x_2^2\} + \frac{\sqrt{2\pi}}{8}S(4\sqrt{\frac{2}{\pi}}), 0.2|x_2|\right)$, and $\varphi_3 = 3\cos(3x_3) + 4\sin(5x_3)$; finally, (III) observe data as the linear combination of drawn SSVs $y = \varphi_1 + \varphi_2 + \varphi_3$.

The summary statistics of the *ground truth*, data generating process (DGP) attribution of features are depicted in the top row of Figure 3. We observe input-output pairs, i.e., $\mathbb{D} = \{\boldsymbol{x}_i, y_i\}_{i=1}^D$, generated from the DGP above, and use them for training PSI. Recall that $\varphi$ are latent variables, *not available in training*.

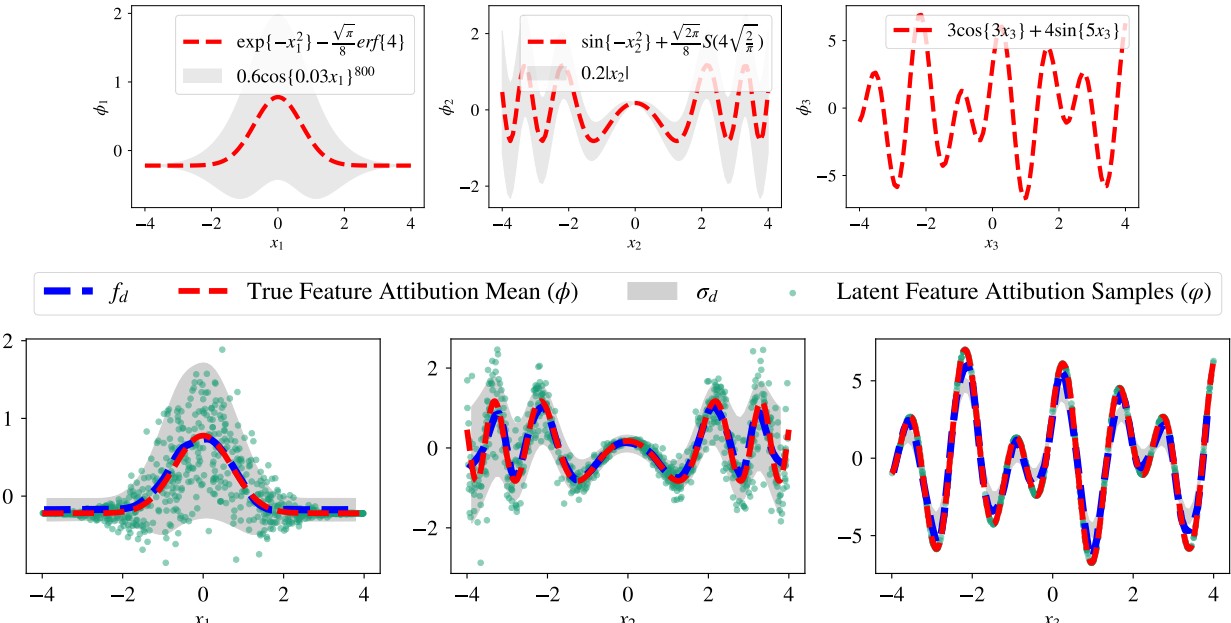

Figure 3: Ground truth feature attributions (top) and the attributions learned by PSI (bottom). Gray areas represent **true** (top) and **learned** (bottom) heteroscedastic uncertainty functions, both of which are two standard deviations away from the mean statistics. The learned PSI mean $f_d$ closely aligns with the true data-generating feature attributions, while PSI' learned $\sigma_d$ accurately captures the underlying attribution uncertainty. Notably, latent SSV samples drawn from the true data generating functions (*not observed during training* and shown in green) fall within two standard deviations of PSI's credible intervals, indicating well-calibrated feature attribution uncertainty.

We showcase in the bottom row of Figure 3 PSI's learned variational distributions over latent $\varphi$ values, i.e., $\mathrm{Q}_{shap_d}(\varphi \mid \boldsymbol{x})$. Each learned $f_d(\boldsymbol{x})$ closely aligns with the mean of the true data-generating feature attributions, while the learned uncertainty $\sigma_d(\boldsymbol{x})$ accurately captures the underlying input-conditional attribution noise.

In particular, we highlight that PSI successfully:

1. **Captures the true underlying heteroscedastic uncertainty** —approximating noise functions $0.6\cos{(0.03x_1)}^{800}$ for $f_1$, $0.2|x_2|$ for $f_2$, and near-zero variance for $f_3$— in alignment with the true data patterns.

2. **Covers the latent, random SSV samples** $\varphi$ within credible intervals of two standard deviations, indicating well-calibrated uncertainty estimates. These latent SSV samples *are not observed*, and are overlaid over the learned summary statistics of Figure 3 solely for illustrative purposes.

## 4.2 On the Benefits of Feature Attribution Decisions based on Credible Intervals

We hereby illustrate the significance of considering distributional Shapley attributions for model explainability, i.e., the added benefits of modeling and learning Shapley distributions for nuanced feature attribution understanding. With access to a variational Shapley distribution $Q_{shap_i}$, uncertainty quantification can be used to probabilistically identify informative features, by considering credible attribution intervals.

We define a probabilistic Shapley attribution as $att_d(z, \boldsymbol{x}) = f_d(\boldsymbol{x}) + z\sigma_d(\boldsymbol{x})$, where we can elucidate attributions based not only on the mean Shapley value, but considering full attribution probabilities: i.e., we can move away from the expected attribution and consider the attribution probability covered by $z$ standard deviations.

There are two hypotheses that motivate the full probabilistic characterization of feature attributions: ($i$) the conditional mean statistic can often relate (and be limited) to *spurious correlations* that explain average attributions of observed data, and ($ii$) irrelevant regions of input feature space —that do not contribute to the observed model outputs— act as *dummy attributions*, and hence tend to exhibit *low attribution uncertainty*. Hence, the rationale for making decisions with higher posterior probabilities is that dummy features will have both *negligible mean and negligible variance*. By increasing the probabilistic attribution threshold, we move beyond average attributions, shifting the focus of explanations to high-confidence attribution regions that better explain the model's decisions.

In what follows, we first assess how feature attribution uncertainty enhances the explainability of image recognition tasks. Then, we showcase how probabilistic ranking of feature attributions reveals subtleties on which inputs most critically influence a model's prediction, aiding subsequent decision-making.

**Localization of relevant image regions on MNIST.** To apply PSI with the proposed MENN architecture to image data, we convert MNIST digit images into a tabular format by dividing each image into $D' \times D'$ non-overlapping patches, where $D' < D$, and $D$ is the original image height/width. Each patch is *individually* mapped to a scalar value, producing a latent representation that serves as input to a decoder reconstructing the original image.

We use this representation to learn the PSI model, i.e., $\boldsymbol{x}$ is the encoded image representation, $y$ is the per-image latent logit random variable, and $\mathbf{1}$ is the observed label. We train separate models on four of the MNIST digits (0, 2, 4, and 8), using one versus all binary classifiers with Bernoulli likelihoods.[2]

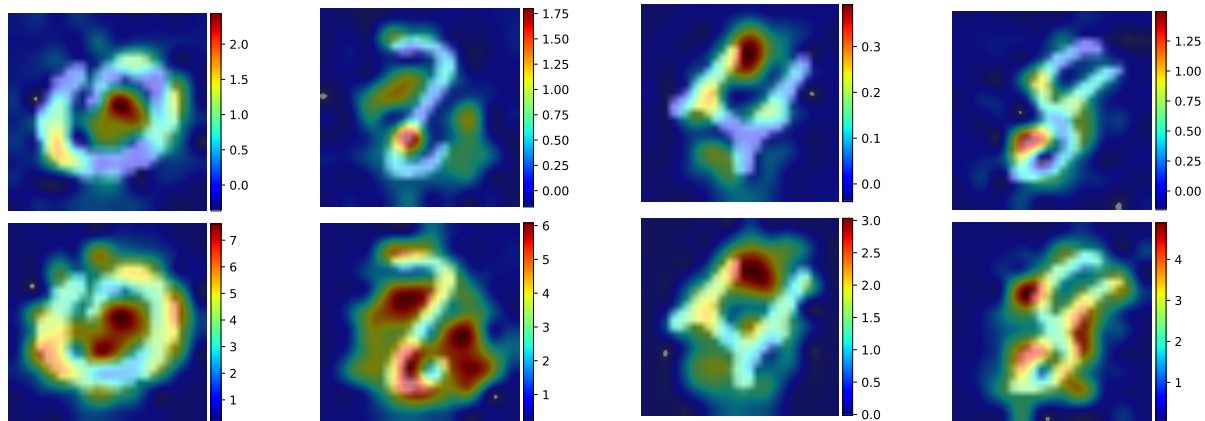

Figure 4: MNIST digit localization according to $att_d(z, \boldsymbol{x})$ with respect to varying $z$-values. (**top**) The 50% credible attribution ($z = 0$), that corresponds to standard Shapley values, often results in coarse or low-resolution attributions. In contrast, (**bottom**) using 98% confidence intervals ($z = 2$) yields more informative feature attributions localized around the full digit of interest and its key traces.

---

[2]Extension of PSI to the multinomial likelihood is left as future work.

In Figure 4, we illustrate how decisions based on probabilistic Shapley attributions $att_d(z, \boldsymbol{x})$, for varying $z$, enhances explainability. In the top row, we show how relying solely on mean attributions, i.e., standard Shapley values, can lead to low information granularity. For instance, the mean attribution for digit 0 emphasizes the hollow center significantly, rather than the digit's outline, reflecting a spurious pattern. Similarly, for digit 4, the mean attribution highlights the empty space between the vertical and diagonal strokes. In contrast, when using $att_d(z = 2, \boldsymbol{x})$, the attribution map more effectively highlights the digit regions where the actual number is displayed, demonstrating improved localization of relevant features. We emphasize that dummy features, i.e., regions that do not include digit strokes, have zero attribution for both $z = 0$ and $z = 2$ (illustrated in dark-blue color in Figure 4).

We quantify the qualitative explainability improvement described above with quantitative results presented in Figure 5, for varying decision thresholds, from $z = 0$ to $z = 8$. We first standardize the attribution scores, $\widetilde{att_d}(z, \boldsymbol{x}) = \frac{att_d(z,\boldsymbol{x}) - \inf_{d \in [D]} att_d(z,\boldsymbol{x})}{\sup_{d \in [D]} att_d(z,\boldsymbol{x}) - \inf_{d \in [D]} att_d(z,\boldsymbol{x})}$, to ensure that probabilistic Shapley attributions are between 0 and 1, hence enabling consistent comparison across instances. We measure the precision ($PR = \frac{TP}{TP+FP}$), recall ($RCL = \frac{TP}{TP+FN}$), and F-metrics ($F = 2\frac{RCL \times PR}{RCL+PR}$) according to the definitions that follow.

We define true positives as $TP = \mathbb{E}_{\boldsymbol{x} \sim \mathbb{D}} \left\{ \mathbb{E}_{d \sim U(1, D')} \left\{ x_d * \widetilde{att_d}(z, \boldsymbol{x}) | \boldsymbol{x}, x_d \neq 0 \right\} \right\}$, where high TP values correspond to when the attribution score is high in regions where the digit is present (i.e., $x_d = 1$); false positives as $FP = \mathbb{E}_{\boldsymbol{x} \sim \mathbb{D}} \left\{ \mathbb{E}_{d \sim U(1, D')} \left\{ (1 - x_d) * \widetilde{att_d}(z, \boldsymbol{x}) | \boldsymbol{x}, x_d = 0 \right\} \right\}$, with high values arising when the attribution score is high in regions without a digit (i.e., $x_d = 0$); and false negatives as $FN = \mathbb{E}_{\boldsymbol{x} \sim \mathbb{D}} \left\{ \mathbb{E}_{d \sim U(1, D')} \left\{ x_d * (1 - \widetilde{att_d}(z, \boldsymbol{x})) | \boldsymbol{x}, x_d \neq 0 \right\} \right\}$, occurring when the attribution score is low in regions where a digit actually exists.

Results in Figure 5 demonstrate that incorporating more attribution mass leads to better localization and explainability, with saturation beyond the $z = 2$ (98% confidence) case.

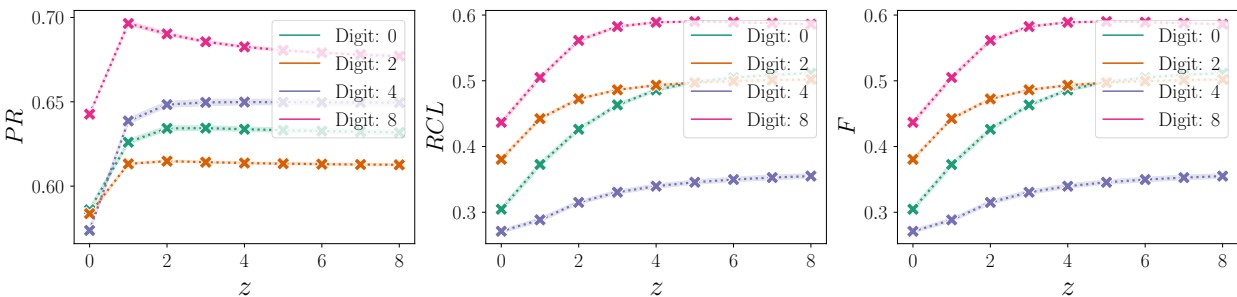

Figure 5: Precision, recall, and F-metrics for varying $z$ value. We observe that for digits 0, 2, and 4 precision increases with $z$ and plateaus around/after $z = 2$. For digit 8, PSI loses precision for $z > 2$, indicating that the model assigns attributions to regions that do not include the digit. We conclude that incorporation of regions outside of mean attributions ($f_d = \phi_d$, with $z = 0$) leads to better localization and interpretability.

**Probabilistic Ranking of Feature Attributions.** Knowing the ranking of feature attributions of a model reveals which inputs most strongly influence the model's predictions, aiding interpretation and decision-making.

We showcase here the use of probabilistic Shapley attribution rankings on a task of clinical relevance. Specifically, we train PSI on the ICU mortality dataset of (Meredith et al., 2020), which contains anonymized electronic health records from ICU patients (including demographics, vital signs, laboratory test results, and interventions over the first 24 hours of admission) for the task of predicting mortality.

In Figure 6, we illustrate how mean attribution-based rankings ($z = 0$, on the left) can differ significantly from worst-case scenarios ($z = 2$, on the right).

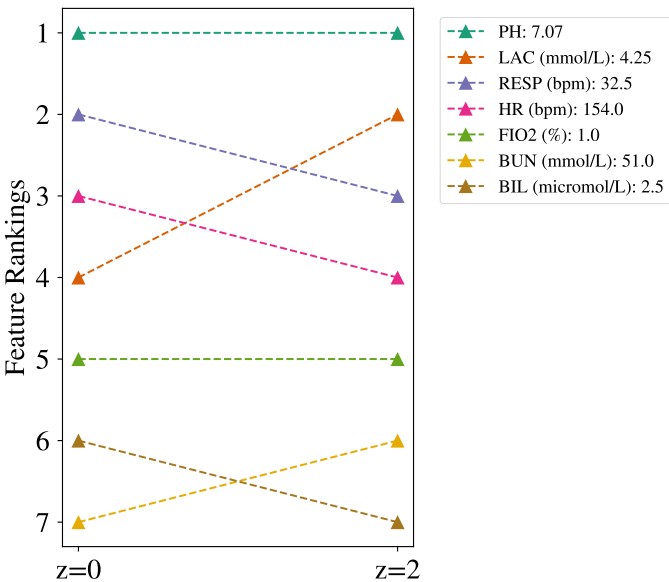

Figure 6: Probabilistic ranking of measurements for predicting ICU mortality of a patient, with a higher ranking indicating higher feature attribution to mortality. Ranking input feature importance varies with respect to the considered credible interval, highlighting different insights on their significance.

In particular, from the mean feature attribution ranks in Figure 6, one would conclude that respiratory and heart rates (RESP and HR) are, on average, more influential than lactate (LAC). However, the uncertainty around LAC's attribution is significantly higher, suggesting that in certain high-risk cases, LAC may be one of the most critical features. Indeed, as explained by a clinical expert, a lactate level of 4.25 mmol/L reflects severe tissue hypoperfusion and is strongly associated with life-threatening conditions like sepsis or shock, whereas a respiratory rate of 32.5 bpm, while elevated, typically indicates compensatory stress and is less directly tied to mortality. In other words, RESP and HR are typically indicative of risk rather than a direct marker of a life-threatening pathology. In contrast, LAC, which is at the top of the rank when considering feature attribution probabilities, is a direct indicator of critical outcomes.

### 4.3   PSI: efficient and accurate data-marginal computations via MENN

With this ablation study, we demonstrate the benefits of the MENN architecture proposed in Section 3.4.1 for efficiently and accurately computing data-marginals of variable input-length.

We provide comparative predictive performance results for synthetic regression tasks in Table 1 (see Appendix F.1 for their corresponding DGPs), by comparing PSI's performance when using MENN or a standard feedforward neural network (FFNN). Precisely, we replicate the FFNN-based procedure of Jethani et al. (2021), where variable-length input modeling is achieved by replacing removed features with a baseline value outside the data support, based on inputting the FFNN with a concatenation of data and the feature removal values $R$. We train both alternatives using the PSI objective, and compare their predictive —Root Mean Squared Error (RMSE)— performance.

We observe in Table 1 that modeling variable-length inputs using the MENN architecture leads to better predictive performance, across different scaling factors of $\hat{D}_{SHAP}$, grouped together as $\beta' = D\beta/2$.

Additionally, we demonstrate how data samples drawn from the MENN-based PSI procedure are closer to the true data distribution, i.e., it provides a more accurate description of the observed data. To that end, we measure Jeffreys-Divergence between empirical samples drawn from PSI and the original, synthetic data generating process —we refer to Appendix F.6 for details on how we compute these divergences.

Table 1: PSI's predictive performace (RMSE) results when using different model architectures, for various $\beta' = D\beta/2$ values in $\mathcal{V}_{reg}$. The best results are shown in **bold**. The proposed masked network (MENN) outperforms the widely used feedforward (FFNN) baseline on all datasets (see Appendix F.1 for their DGPs).

| Net | $\beta'$ | synth1 | synth2 | synth3 | synth4 | synth5 |
|---|---|---|---|---|---|---|
| MENN | 0.001 | 0.605 | **0.004** | 0.125 | 0.669 | 0.419 |
| | 0.01 | 0.611 | **0.004** | **0.120** | 0.669 | **0.416** |
| | 0.1 | 0.595 | **0.004** | 0.221 | 0.673 | 0.417 |
| | 1 | **0.582** | **0.004** | 1.118 | **0.579** | 0.446 |
| FFNN | 0.001 | 0.642 | 0.071 | 0.368 | 0.689 | 0.433 |
| | 0.01 | 0.640 | 0.128 | 0.234 | 0.677 | 0.435 |
| | 0.1 | 0.624 | 0.103 | 0.174 | 0.674 | 0.431 |
| | 1 | 0.599 | 0.186 | 0.897 | 0.660 | 0.477 |

In Figure 7, we clearly observe that the MENN architecture leads to better data-marginal density modeling.

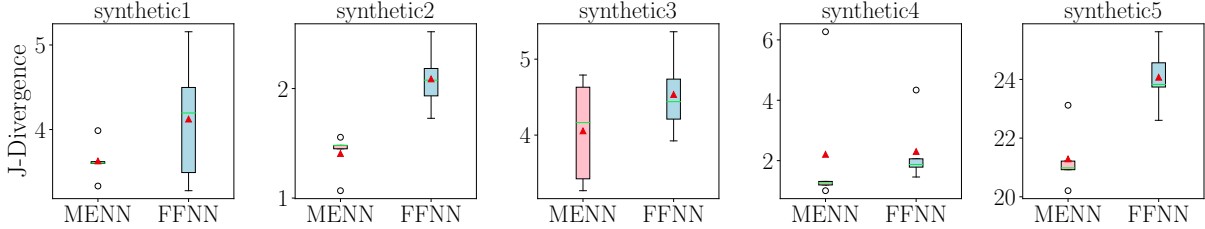

Figure 7: Jeffreys (J)-Divergence of data drawn from PSI to the observed empirical data, per considered architecture. ▲ is the mean and − is the median of results across folds, error bars denote the 1.5 interquartile range. We observe that data simulated using MENN leads to values closer to the observed data distribution, as indicated by lower divergences.

## 4.4 Predictive Performance Results

In this section, we perform 5-fold cross validation to assess, in addition to PSI's benefits on explainability, its predictive performance.

Table 2: Averaged results (see Appendix F.5 for their negligible standard errors) on 8 real-world datasets. We report RMSE for regression (purple) and PR-AUC for classification (pink) datasets. We indicate best overall performance in **bold**, with best performing explainable models in blue, where lower RMSE and higher PR-AUC scores are better.

| Data | Interpretable | | | Black-box | | | |
|---|---|---|---|---|---|---|---|
| | **PSI** | LIN | EBM | RF | LGBM | XGB | DNN |
| PKSN | **0.026** | 0.867 | 0.195 | 0.040 | 0.072 | 0.063 | 0.111 |
| MED | **0.447** | 0.608 | **0.447** | 0.448 | **0.447** | 0.454 | 0.467 |
| BIKE | 0.019 | 0.518 | 0.038 | **0.008** | 0.017 | 0.013 | 0.019 |
| GAS | 0.269 | 60.93 | 0.153 | 0.151 | 0.152 | **0.150** | 0.256 |
| FICO | 0.771 | 0.766 | 0.771 | 0.770 | **0.773** | **0.773** | 0.771 |
| SPAM | 0.977 | 0.946 | 0.977 | **0.983** | 0.982 | 0.981 | 0.967 |
| ICU | 0.872 | 0.851 | 0.872 | 0.869 | **0.874** | **0.874** | 0.860 |
| CENS | 0.803 | 0.768 | 0.825 | 0.800 | 0.830 | **0.831** | 0.788 |

Table 2 presents a comparison between PSI and several established baselines in four real-world regression and classification datasets —details on the baselines and the datasets are provided in Appendix F.3 and F.2, respectively. Overall, PSI achieves predictive performance in par with not only state-of-the-art interpretable methods, but strong black-box models such as Random Forests, XG-Boost and deep neural networks.[3]

## 5 Discussion and Limitations

PSI provides a probabilistic framework for feature attribution via generative modeling of Shapley value-centered latent random variables, with input-conditional Shapley priors. Inference over these latent variables of interest for learning feature attribution distributions is enabled by efficient variational inference. We acknowledge that PSI does not explicitly enforce the $f_d = \phi_d$ constraint, instead relying on regularization through the $D_{SHAP}$ term in the variational objective. Diagnostic tools, as described in Remark 3.3, assist practitioners in evaluating this alignment; as validated by the presented empirical performance in different case studies. However, we recall that the condition $\mathbb{E}_{\boldsymbol{x}}\{f_d(\boldsymbol{x})\} \to 0$ is necessary but not sufficient for guaranteeing $f_d = \phi_d$. Looking ahead, promising directions for extending PSI include support for multinomial likelihoods and the incorporation of causal feature attributions.

## 6 Conclusion

We presented Probabilistic Shapley inference (PSI), a probabilistic framework for feature attribution that models feature attribution values as latent random variables with input-conditional priors, centered around Shapley values imposed by the data generating process, linking attribution uncertainty to predictive uncertainty. We present a revised variational inference framework that efficiently, via stochastic optimization, enables simultaneous inference over the latent Shapley values of a flexible (e.g., neural network-based) predictive model. Empirical results across synthetic and real-world datasets demonstrate not only the explainability benefits of the probabilistic treatment of feature attributions intrinsic to PSI, but its flexibility in accurately describing data marginals, feature attributions and their uncertainty, while providing state-of-the-art predictive performance.

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

# A  Proof of Proposition 3.2

The proof of Proposition 3.2 follows:

For $Q_{shap_d}(\varphi_d \mid \boldsymbol{x}) = \mathcal{N}(f_d(\boldsymbol{x}), \sigma_d^2(\boldsymbol{x}))$ and any function $h$, the **efficiency** property of Shapley values guarantees that $\mathbb{E}_{Q_{shap_i}} \left\{ h \left( \phi_0 + \sum_{d=1}^{D} \varphi_{i,d} \right) \right\}$ is equal to $\mathbb{E}_{P(\Phi_i \mid \boldsymbol{x}_i)} \left\{ h \left( \phi_0 + \sum_{d=1}^{D} \varphi_{i,d} \right) \right\}$.

*Proof.* Showing that $\phi_0 + \sum_{d=1}^{D} \varphi_{i,d}$ is the same random variable under P and Q concludes the proof.

Since, $Q(\varphi_d \mid \boldsymbol{x}) = \mathcal{N} \left( f_d(\boldsymbol{x}), \sigma_d^2(\boldsymbol{x}) \right)$, $\phi_0 + \sum_{d=1}^{D} \varphi_{i,d} \sim \mathcal{N} \left( \phi_0 + \sum_{d}^{D} f_d(\boldsymbol{x}), \sum_{d}^{D} \sigma_d^2(\boldsymbol{x}) \right)$, under Q, and

given $P(\varphi_d \mid \boldsymbol{x}) = \mathcal{N} \left( \sum_{S \subseteq [D] \setminus \{d\}} P(S) \left( f(\boldsymbol{x}_{S \cup \{d\}}) - f(\boldsymbol{x}_s) \right), \sigma_d^2(\boldsymbol{x}) \right)$,

$\phi_0 + \sum_{d=1}^{D} \varphi_{i,d} \sim \mathcal{N} \left( \phi_0 + \sum_{d}^{D} f_d(\boldsymbol{x}), \sum_{d}^{D} \sigma_d^2(\boldsymbol{x}) \right)$, under P due to the efficiency property.

Since sum over $\varphi$ have the same distribution under P and Q, the expected values with respect to these distributions are the same. $\qquad\square$

# B  Details on PSI's Variational Evidence Lower-Bounds

We describe here the procedures to bound data marginals $\log P(y \mid \boldsymbol{x})$ and $\log P(\mathbf{1} \mid \boldsymbol{x})$ for PSI on regression and classification tasks.

**Regression.**  By direct application of Proposition 3.2, we write

$$\mathcal{L}_{reg}^{ELBO} \leq \mathcal{V}_{reg} = \sum_{i=1}^{N} \log P(y_i \mid \boldsymbol{x}_i) - \beta D_{\mathrm{KL}} \left( Q_{shap_i} \| P(\Phi_i \mid \boldsymbol{x}_i) \right) \leq \mathcal{L}_{reg} , \tag{25}$$

which follows from $\mathcal{L}_{reg} = \sum_{i=1}^{N} \log P(y_i \mid \boldsymbol{x}_i)$ given that $D_{\mathrm{KL}} \left( Q_{shap_i} \| P(\Phi_i \mid \boldsymbol{x}_i) \right) \geq 0$.

The inequality is valid for $0 \leq \beta \leq 1$, with $\mathcal{V}_{reg} = \mathcal{L}_{reg}$ for $\beta = 0$. Note that the lower-bound over the marginal log-likelihood is valid $\forall \beta \geq 0$, and the upper bound $\beta \leq 1$ is only necessary to keep the $\mathcal{V}_{reg} \geq \mathcal{L}_{reg}^{ELBO}$ constraint, with equality when $\beta = 1$.

**Classification.**  For binary classification, we start by writing the ELBO as

$$\mathcal{L}_{class}^{ELBO} = \sum_{i=1}^{N} \mathbb{E}_{Q_{logit_i} Q_{shap_i}} \left\{ \log P(\mathbf{1}_i \mid y_i) P(y_i \mid \Phi_i) P(\Phi_i \mid \boldsymbol{x}_i) - \log Q_{logit_i} Q_{shap_i} \right\} \tag{26}$$

$$= \sum_{i=1}^{N} \mathbb{E}_{Q_{logit_i}} \left\{ \log P(\mathbf{1}_i \mid y_i) - \log Q_{logit_i} + \mathbb{E}_{Q_{shap_i}} \left\{ \log P(y_i \mid \Phi_i) - D_{\mathrm{KL}} \left( Q_{shap_i} \| P(\Phi_i \mid \boldsymbol{x}_i) \right) \right\} \right\} \tag{27}$$

$$= \sum_{i=1}^{N} \mathbb{E}_{Q_{logit_i}} \left\{ \log P(\mathbf{1}_i \mid y_i) - \log Q_{logit_i} + \mathcal{L}_{reg_i}^{ELBO} \right\} \tag{28}$$

$$= \sum_{i=1}^{N} \mathbb{E}_{Q_{logit_i}} \left\{ \log P(\mathbf{1}_i \mid y_i) + \mathcal{L}_{reg_i}^{ELBO} \right\} + \mathcal{H}(Q_{logit_i}), \tag{29}$$

where $\mathcal{H}(Q_{logit_i})$ is the entropy of the variational logit distribution $Q_{logit_i}$. Since $\mathcal{L}_{reg_i}^{ELBO} \leq \mathcal{V}_{reg_i}$

$$\mathcal{L}_{class}^{ELBO} \leq \mathcal{V}_{class} = \sum_{i=1}^{N} \mathbb{E}_{Q_{logit_i}} \left\{ \log P(\mathbf{1}_i \mid y_i) + \mathcal{V}_{reg_i} \right\} + \mathcal{H}(Q_{logit_i}) \leq \mathcal{L}_{class} \; , \tag{30}$$

$$\mathcal{V}_{class} = \sum_{i=1}^{N} \mathbb{E}_{Q_{logit_i}} \left\{ \log P(\mathbf{1}_i \mid y_i) + \log P(y_i \mid \boldsymbol{x}_i) \right\} + \mathcal{H}(Q_{logit_i}) - \beta D_{\mathrm{KL}} \left( Q_{shap_i} \| P(\Phi_i \mid \boldsymbol{x}_i) \right) \leq \mathcal{L}_{class} \; . \tag{31}$$

An alternative view of this objective is to lower-bound the marginal log-likelihood by first collapsing the random variables $\Phi$, and subsequently adding the $D_{SHAP} \geq 0$ term to the objective.

## C  Feature Removal and the Subset Marginal Constraint During Optimization

We here show that for PSI's variational regression and classification loss functions $\mathcal{V}_{reg}$ and $\mathcal{V}_{class}$, respectively, introducing missing values during training satisfies the integral constraint:

$$f(\boldsymbol{x}_s) = \int f(\boldsymbol{x}) \, \mathrm{dP}(\boldsymbol{x} \mid \boldsymbol{x}_s) \; . \tag{32}$$

We first clarify the notation used in the proofs, where we denote the true data generating distribution with $P^{\star}(\cdot)$ and use parameterized distributions to denote the PSI model and variational families:

- $P^{\star}(\cdot)$ True data generating distribution (analytical form is often unknown) —notice, without explicit parametric dependency

- $P(\cdot; \theta)$: Model distribution, with parameters $\theta$ —which for notation simplicity, was suppressed in the main text and denoted as $P(\cdot)$

- $Q(\cdot; \lambda)$: Variational distribution, with variational parameters $\lambda$ —which for notation simplicity, was suppressed in the main text and denoted as $Q(\cdot)$

- $\boldsymbol{x}_s$: Variable length input features, i.e., a subset of features fed to the neural network, where features $d$ not in $s$ (i.e., $d \notin s$) are replaced by their corresponding baseline vector $\boldsymbol{b}_d$, as described in Section 3.4.

We now proceed with the proofs, where for regression tasks, derivation is straightforward, as PSI model learning is carried out by optimizing the marginal log-likelihood, which is a proper scoring rule. For classification, we show that the integral constraint is still satisfied under two assumptions detailed below.

**Regression.**  The proof relies on showing that maximizing $\mathcal{V}_{reg}$ as in Equation 16 by introducing missing values during training is equivalent to minimizing the following, averaged linear combination of divergences:

$$\mathbb{E}_{P^{\star}(\boldsymbol{x})} \left\{ D_{\mathrm{KL}} \left( P^{\star}(y \mid \boldsymbol{x}) \| P(y \mid \boldsymbol{x}_s; \theta) \right) + \beta D_{\mathrm{KL}} \left( Q(\Phi \mid \boldsymbol{x}_s; \lambda) \| P(\Phi \mid \boldsymbol{x}_s; \theta) \right) \right\} . \tag{33}$$

First, we show that Equation 33 is a lower bound of $-\mathbb{E}_{P(\boldsymbol{x})} \left\{ \mathcal{V}_{reg_i} \right\}$, by a constant factor:

$$\mathbb{E}_{P^{\star}(\boldsymbol{x})} \left\{ D_{\mathrm{KL}} \left( P^{\star}(y \mid \boldsymbol{x}) \| P(y \mid \boldsymbol{x}_s; \theta) \right) + \beta D_{\mathrm{KL}} \left( Q(\Phi \mid \boldsymbol{x}_s; \lambda) \| P(\Phi \mid \boldsymbol{x}_s; \theta) \right) \right\} \tag{34}$$

$$= \mathbb{E}_{P^{\star}(\boldsymbol{x}) P^{\star}(y \mid \boldsymbol{x})} \left\{ \log \left( \frac{P^{\star}(y \mid \boldsymbol{x})}{P(y \mid \boldsymbol{x}_s; \theta)} \right) + \beta D_{\mathrm{KL}} \left( Q(\Phi \mid \boldsymbol{x}_s; \lambda) \| P(\Phi \mid \boldsymbol{x}_s; \theta) \right) \right\} \tag{35}$$

$$= \underbrace{-\mathbb{E}_{P^{\star}(\boldsymbol{x}) P^{\star}(y \mid \boldsymbol{x})} \left\{ \log P(y \mid \boldsymbol{x}_s; \theta) - \beta D_{\mathrm{KL}} \left( Q(\Phi \mid \boldsymbol{x}_s; \lambda) \| P(\Phi \mid \boldsymbol{x}_s; \theta) \right) \right\}}_{-\mathbb{E}_{P^{\star}(\boldsymbol{x})} \left\{ \mathcal{V}_{reg_i} \right\}} + \underbrace{\mathbb{E}_{P^{\star}(\boldsymbol{x}) P^{\star}(y \mid \boldsymbol{x})} \left\{ \log P^{\star}(y \mid \boldsymbol{x}) \right\}}_{C_1}$$

$$\tag{36}$$

where $C_1 = \mathbb{E}_{\mathrm{P}^\star(\boldsymbol{x})\mathrm{P}^\star(y|\boldsymbol{x})}\{\log \mathrm{P}^\star(y \mid \boldsymbol{x})\}$ is only dependent on the data distribution, and hence a constant with respect to model or variational parameters. Moving forward, we recall that $-\frac{1}{N}\mathcal{V}_{reg} \xrightarrow{p} -\mathbb{E}_{\mathrm{P}^\star(\boldsymbol{x})}\{\mathcal{V}_{reg_i}\}$, as $N \to \infty$, and we elaborate on

$$-\frac{1}{N}\mathcal{V}_{reg} + C_1 = \mathbb{E}_{\mathrm{P}^\star(\boldsymbol{x})\mathrm{P}^\star(y|\boldsymbol{x})}\left\{\log \frac{\mathrm{P}^\star(y \mid \boldsymbol{x})}{\mathrm{P}(y \mid \boldsymbol{x}_s; \theta)} + \beta D_{\mathrm{KL}}\left(\mathrm{Q}(\Phi \mid \boldsymbol{x}_s; \lambda)\|\mathrm{P}(\Phi \mid \boldsymbol{x}_s; \theta)\right)\right\}, \qquad (37)$$

by dividing and multiplying by the data feature (covariate) true distribution $\mathrm{P}^\star(\boldsymbol{x})$ the log-ratio within the first term of the right hand side:

$$-\frac{1}{N}\mathcal{V}_{reg} + C_1 = \mathbb{E}_{\mathrm{P}^\star(\boldsymbol{x})\mathrm{P}^\star(y|\boldsymbol{x})}\left\{\log \frac{\mathrm{P}^\star(y \mid \boldsymbol{x})\mathrm{P}^\star(\boldsymbol{x})}{\mathrm{P}(y \mid \boldsymbol{x}_s; \theta)\mathrm{P}^\star(\boldsymbol{x})} + \beta D_{\mathrm{KL}}\left(\mathrm{Q}(\Phi \mid \boldsymbol{x}_s; \lambda)\|\mathrm{P}(\Phi \mid \boldsymbol{x}_s; \theta)\right)\right\}. \qquad (38)$$

Rearranging the terms according to Bayes' rule, we have

$$-\frac{1}{N}\mathcal{V}_{reg} + C_1 = \mathbb{E}_{\mathrm{P}^\star(\boldsymbol{x})\mathrm{P}^\star(y|\boldsymbol{x})}\left\{\log \frac{\mathrm{P}^\star(y \mid \boldsymbol{x}_s)\mathrm{P}^\star(\boldsymbol{x}_{-s} \mid \boldsymbol{x}_s, y)\mathrm{P}^\star(\boldsymbol{x}_s)}{\mathrm{P}(y \mid \boldsymbol{x}_s; \theta)\mathrm{P}^\star(\boldsymbol{x}_{-s} \mid \boldsymbol{x}_s)\mathrm{P}^\star(\boldsymbol{x}_s)} + \beta D_{\mathrm{KL}}\left(\mathrm{Q}(\Phi \mid \boldsymbol{x}_s; \lambda)\|\mathrm{P}(\Phi \mid \boldsymbol{x}_s; \theta)\right)\right\}. \qquad (39)$$

We group expectations in terms of KL divergences to obtain:

$$-\frac{1}{N}\mathcal{V}_{reg} + C_1 = \mathbb{E}_{\mathrm{P}^\star(\boldsymbol{x}_s)}\left\{\underbrace{D_{\mathrm{KL}}\left(\mathrm{P}^\star(y \mid \boldsymbol{x}_s)\|\mathrm{P}(y \mid \boldsymbol{x}_s; \theta)\right)}_{(\mathrm{I})}\right\}$$

$$+ \mathbb{E}_{\mathrm{P}^\star(\boldsymbol{x}_s)}\left\{\underbrace{\beta D_{\mathrm{KL}}\left(\mathrm{Q}(\Phi \mid \boldsymbol{x}_s; \lambda)\|\mathrm{P}(\Phi \mid \boldsymbol{x}_s; \theta)\right)}_{(\mathrm{II})}\right\}$$

$$+ \underbrace{\mathbb{E}_{\mathrm{P}^\star(\boldsymbol{x})\mathrm{P}^\star(y|\boldsymbol{x})}\left\{D_{\mathrm{KL}}\left(\mathrm{P}^\star(\boldsymbol{x}_{-s} \mid \boldsymbol{x}_s, y)\|\mathrm{P}^\star(\boldsymbol{x}_{-s} \mid \boldsymbol{x}_s)\right)\right\}}_{C_2}, \qquad (40)$$

where we notice that $C_2$ is a constant that does not depend on the parametric PSI and variational models, but on the true data distribution (e.g., if $\boldsymbol{x}_{-s} \perp y \mid \boldsymbol{x}_s$, then $C_2 = 0$),

Equation 40 is minimized when (I) the model marginal is equal to the data distribution's marginal, i.e., $\mathrm{P}^\star(y \mid \boldsymbol{x}_s) = \mathrm{P}(y \mid \boldsymbol{x}_s; \theta)$ and (II) the Shapley Kullback-Leibler divergence is 0, i.e., $\mathrm{Q}(\Phi \mid \boldsymbol{x}_s; \lambda) = \mathrm{P}(\Phi \mid \boldsymbol{x}_s; \theta)$. Therefore, for a sufficiently flexible conditional model, the model marginal likelihood will converge to the data's marginal density, while the variational distribution will capture the (computationally expensive) Shapley prior.

**Classification.** The difference between PSI's regression and classification objectives is that in the latter, we also approximate the (in this case, latent) $y$ values via a variational distribution, which defines the logits of observed labels $\mathbf{1}_i$. In what follows, we assume that there exists a data generating distribution under which $\mathrm{P}^\star(\mathbf{1} \mid \boldsymbol{x}) = \int \mathrm{P}^\star(\mathbf{1} \mid y)\mathrm{P}^\star(y \mid \boldsymbol{x})\,\mathrm{d}y$. I.e., there exists a data generating $\mathrm{P}^\star(y \mid \boldsymbol{x})$ for which the integral can be computed.

The argument again relies on showing that maximizing $\mathcal{V}_{class}$ as in Equation 18 by introducing missing values during training is equal to minimizing the following divergence:

$$\mathbb{E}_{\mathrm{P}^\star(\boldsymbol{x})}\left\{D_{\mathrm{KL}}\left(\mathrm{P}^\star(\mathbf{1} \mid \boldsymbol{x})\mathrm{Q}(y \mid \boldsymbol{x}_s; \lambda)\|\mathrm{P}(\mathbf{1} \mid y; \theta)\mathrm{P}(y \mid \boldsymbol{x}_s; \theta)\right) + \beta D_{\mathrm{KL}}\left(\mathrm{Q}(\Phi \mid \boldsymbol{x}_s; \lambda)\|\mathrm{P}(\Phi \mid \boldsymbol{x}_s; \theta)\right)\right\}. \qquad (41)$$

We show the above by rewriting it as

$$\mathbb{E}_{\mathrm{P}^\star(\boldsymbol{x})}\left\{D_{\mathrm{KL}}\left(\mathrm{P}^\star(\mathbf{1}\mid\boldsymbol{x})\mathrm{Q}(y\mid\boldsymbol{x}_s;\lambda)\|\mathrm{P}(\mathbf{1}\mid y;\theta)\mathrm{P}(y\mid\boldsymbol{x}_s;\theta)\right)+\beta D_{\mathrm{KL}}\left(\mathrm{Q}(\Phi\mid\boldsymbol{x}_s;\lambda)\|\mathrm{P}(\Phi\mid\boldsymbol{x}_s;\theta)\right)\right\} \tag{42}$$

$$=\mathbb{E}_{\mathrm{P}^\star(\boldsymbol{x})\mathrm{P}^\star(\mathbf{1}\mid\boldsymbol{x})\mathrm{Q}(y\mid\boldsymbol{x}_s;\lambda)}\left\{\log\left(\frac{\mathrm{P}^\star(\mathbf{1}\mid\boldsymbol{x})\mathrm{Q}(y\mid\boldsymbol{x}_s;\lambda)}{\mathrm{P}(\mathbf{1}\mid y;\theta)\mathrm{P}(y\mid\boldsymbol{x}_s;\theta)}\right)+\beta D_{\mathrm{KL}}\left(\mathrm{Q}(\Phi\mid\boldsymbol{x}_s;\lambda)\|\mathrm{P}(\Phi\mid\boldsymbol{x}_s;\theta)\right)\right\}$$

$$=\mathbb{E}_{\mathrm{P}^\star(\boldsymbol{x})\mathrm{P}^\star(\mathbf{1}\mid\boldsymbol{x})\mathrm{Q}(y\mid\boldsymbol{x}_s;\lambda)}\left\{-\log\mathrm{P}(\mathbf{1}\mid y;\theta)+D_{\mathrm{KL}}\left(\mathrm{Q}(y\mid\boldsymbol{x}_s;\lambda)\|\mathrm{P}(y\mid\boldsymbol{x}_s;\theta)\right)\right\} \tag{43}$$

$$+\mathbb{E}_{\mathrm{P}^\star(\boldsymbol{x})\mathrm{P}^\star(\mathbf{1}\mid\boldsymbol{x})\mathrm{Q}(y\mid\boldsymbol{x}_s;\lambda)}\left\{\beta D_{\mathrm{KL}}\left(\mathrm{Q}(\Phi\mid\boldsymbol{x}_s;\lambda)\|\mathrm{P}(\Phi\mid\boldsymbol{x}_s;\theta)\right)\right\}$$

$$+\underbrace{\mathbb{E}_{\mathrm{P}^\star(\boldsymbol{x})\mathrm{P}^\star(\mathbf{1}\mid\boldsymbol{x})}\left\{\log\mathrm{P}^\star(\mathbf{1}\mid\boldsymbol{x})\right\}}_{C_3} \tag{44}$$

where we recall that the outer expectation over $\mathrm{Q}(y\mid\boldsymbol{x}_s;\lambda)$ does not affect the $D_{\mathrm{KL}}\left(\mathrm{Q}(y\mid\boldsymbol{x}_s;\lambda)\|\mathrm{P}(y\mid\boldsymbol{x}_s;\theta)\right)$ term, as this is a constant after its own marginalization of $\mathrm{Q}(y\mid\boldsymbol{x}_s;\lambda)$. The above, where the first 3 terms in Equation 44 form $-\mathbb{E}_{\mathrm{P}^\star(\boldsymbol{x})}\left\{\mathcal{V}_{class}\right\}$, showcases that Equation 41 bounds $-\frac{1}{N}\mathcal{V}_{class}\xrightarrow{p}-\mathbb{E}_{\mathrm{P}^\star(\boldsymbol{x})}\left\{\mathcal{V}_{class_i}\right\}$, as $N\to\infty$, by a constant factor $C_3$.

We then write

$$-\frac{1}{N}\mathcal{V}_{class}+C_3=\mathbb{E}_{\mathrm{P}^\star(\boldsymbol{x})\mathrm{P}^\star(\mathbf{1}\mid\boldsymbol{x})\mathrm{Q}(y\mid\boldsymbol{x}_s;\lambda)}\left\{\log\frac{\mathrm{P}^\star(\mathbf{1}\mid\boldsymbol{x})\mathrm{Q}(y\mid\boldsymbol{x}_s;\lambda)}{\mathrm{P}(\mathbf{1}\mid y;\theta)\mathrm{P}(y\mid\boldsymbol{x}_s;\theta)}+\beta D_{\mathrm{KL}}\left(\mathrm{Q}(\Phi\mid\boldsymbol{x}_s;\lambda)\|\mathrm{P}(\Phi\mid\boldsymbol{x}_s;\theta)\right)\right\}, \tag{45}$$

and multiplying the log-ratio within the first term with $\mathrm{P}^\star(y\mid\boldsymbol{x}_s)$, we have

$$-\frac{1}{N}\mathcal{V}_{class}+C_3=\mathbb{E}_{\mathrm{P}^\star(\boldsymbol{x})\mathrm{P}^\star(\mathbf{1}\mid\boldsymbol{x})\mathrm{Q}(y\mid\boldsymbol{x}_s;\lambda)}\left\{\log\frac{\mathrm{P}^\star(\mathbf{1}\mid\boldsymbol{x})\mathrm{Q}(y\mid\boldsymbol{x}_s;\lambda)\mathrm{P}^\star(y\mid\boldsymbol{x}_s)}{\mathrm{P}(\mathbf{1}\mid y;\theta)\mathrm{P}(y\mid\boldsymbol{x}_s;\theta)\mathrm{P}^\star(y\mid\boldsymbol{x}_s)}\right\}$$

$$+\mathbb{E}_{\mathrm{P}^\star(\boldsymbol{x})\mathrm{P}^\star(\mathbf{1}\mid\boldsymbol{x})\mathrm{Q}(y\mid\boldsymbol{x}_s;\lambda)}\left\{\beta D_{\mathrm{KL}}\left(\mathrm{Q}(\Phi\mid\boldsymbol{x}_s;\lambda)\|\mathrm{P}(\Phi\mid\boldsymbol{x}_s;\theta)\right)\right\}, \tag{46}$$

which can be rearranged as:

$$-\frac{1}{N}\mathcal{V}_{class}+C_3=\mathbb{E}_{\mathrm{P}^\star(\boldsymbol{x})\mathrm{P}^\star(\mathbf{1}\mid\boldsymbol{x})\mathrm{Q}(y\mid\boldsymbol{x}_s;\lambda)}\left\{\log\frac{\mathrm{P}^\star(\mathbf{1}\mid\boldsymbol{x})\mathrm{P}^\star(y\mid\boldsymbol{x}_s)}{\mathrm{P}(\mathbf{1}\mid y;\theta)\mathrm{P}(y\mid\boldsymbol{x}_s;\lambda)}\right\}+D_{\mathrm{KL}}\left(\mathrm{Q}(y\mid\boldsymbol{x}_s;\lambda)\|\mathrm{P}^\star(y\mid\boldsymbol{x}_s)\right)$$

$$+\mathbb{E}_{\mathrm{P}^\star(\boldsymbol{x}_s)}\left\{\beta D_{\mathrm{KL}}\left(\mathrm{Q}(\Phi\mid\boldsymbol{x}_s;\lambda)\|\mathrm{P}(\Phi\mid\boldsymbol{x}_s;\theta)\right)\right\}. \tag{47}$$

Now, as the learning procedure evolves, we assume $D_{\mathrm{KL}}\left(\mathrm{Q}(y\mid\boldsymbol{x}_s;\lambda)\|\mathrm{P}^\star(y\mid\boldsymbol{x}_s)\right)\to 0$:[4]

$$-\frac{1}{N}\mathcal{V}_{class}+C_3=\mathbb{E}_{\mathrm{P}^\star(\boldsymbol{x})\mathrm{P}^\star(\mathbf{1}\mid\boldsymbol{x})\mathrm{P}^\star(y\mid\boldsymbol{x}_s)}\left\{\log\frac{\mathrm{P}^\star(\mathbf{1}\mid\boldsymbol{x})\mathrm{P}^\star(y\mid\boldsymbol{x}_s)}{\mathrm{P}(\mathbf{1}\mid y;\theta)\mathrm{P}(y\mid\boldsymbol{x}_s;\theta)}\right\}$$

$$+\mathbb{E}_{\mathrm{P}^\star(\boldsymbol{x}_s)}\left\{\beta D_{\mathrm{KL}}\left(\mathrm{Q}(\Phi\mid\boldsymbol{x}_s;\lambda)\|\mathrm{P}(\Phi\mid\boldsymbol{x}_s;\theta)\right)\right\} \tag{48}$$

$$=\mathbb{E}_{\mathrm{P}^\star(\boldsymbol{x})\mathrm{P}^\star(\mathbf{1}\mid\boldsymbol{x})\mathrm{P}^\star(y\mid\boldsymbol{x}_s)}\left\{D_{\mathrm{KL}}\left(\mathrm{P}^\star(\mathbf{1}\mid\boldsymbol{x})\|\mathrm{P}(\mathbf{1}\mid y;\theta)\right)\right\}$$

$$+\underbrace{\mathbb{E}_{\mathrm{P}^\star(\boldsymbol{x}_s)}\left\{D_{\mathrm{KL}}\left(\mathrm{P}^\star(y\mid\boldsymbol{x}_s)\|\mathrm{P}(y\mid\boldsymbol{x}_s;\theta)\right)\right\}+\mathbb{E}_{\mathrm{P}^\star(\boldsymbol{x}_s)}\left\{D_{\mathrm{KL}}\left(\mathrm{Q}(\Phi\mid\boldsymbol{x}_s;\lambda)\|\mathrm{P}(\Phi\mid\boldsymbol{x}_s;\theta)\right)\right\}}_{-\frac{1}{N}\mathcal{V}_{reg}+C_1-C_2}$$

$$=\mathbb{E}_{\mathrm{P}^\star(\boldsymbol{x})\mathrm{P}^\star(\mathbf{1}\mid\boldsymbol{x})\mathrm{P}^\star(y\mid\boldsymbol{x}_s)}\left\{\underbrace{D_{\mathrm{KL}}\left(\mathrm{P}^\star(\mathbf{1}\mid\boldsymbol{x})\|\mathrm{P}(\mathbf{1}\mid y;\theta)\right)}_{(\mathrm{I})}-\underbrace{\frac{1}{N}\mathcal{V}_{reg}+C_1-C_2}_{(\mathrm{II})}\right\} \tag{49}$$

Namely, under the assumption that the variational posterior is close to the data-generating posterior and is stable, PSI's classification loss $\mathcal{V}_{class}$: (I) encourages learning latent logit marginals, and (II) partitions their influence to their corresponding Shapley feature attributions via $\mathcal{V}_{reg}$ (which includes $D_{SHAP}$).

---

[4]An assumption based on the universal approximation properties of neural networks.

**Conclusion.** Under both losses, as learning proceeds with $\mathrm{P}(y \mid \boldsymbol{x}_s; \theta) \xrightarrow{d} \mathrm{P}^\star(y \mid \boldsymbol{x}_s)$, we can write:

$$f(\boldsymbol{x}_s) + \phi_0 = \mathbb{E}_{\mathrm{P}(y \mid \boldsymbol{x}_s; \theta)} \{y\} = \mathbb{E}_{\mathrm{P}(y \mid \boldsymbol{x}; \theta) \mathrm{P}^\star(\boldsymbol{x} \mid \boldsymbol{x}_s)} \{y\} = \int f(\boldsymbol{x}) \, \mathrm{dP}^\star(\boldsymbol{x} \mid \boldsymbol{x}_s) + \phi_0. \tag{50}$$

Hence, $f(\boldsymbol{x}_s) = \int f(\boldsymbol{x}) \, \mathrm{dP}^\star(\boldsymbol{x} \mid \boldsymbol{x}_s)$, concluding that, under reasonable learning assumptions, randomly removing feature dimensions during training meets the integral constraint.

## D   Stochastic estimation of the Shapley Kullback-Leibler Divergence

In this section, we characterize the stochastic estimator of the Shapley Kullback-Leibler divergence and demonstrate that the gradients of the proposed stochastic estimator point in the same direction as the exact one's, in expectation.

We start by recalling the definition of the proposed stochastic estimator $\hat{D}_{SHAP}$:

$$\hat{D}_{SHAP} = D \frac{\left| \left( \underbrace{\frac{\sum_{k=1}^{K_1} f(\boldsymbol{x}_{s_{1,k} \cup d}) - f(\boldsymbol{x}_{s_{1,k}})}{K_1} - f_d(\boldsymbol{x})}_{\text{term 1}} \right) \left( \underbrace{\frac{\sum_{k=1}^{K_2} f(\boldsymbol{x}_{s_{2,k} \cup d}) - f(\boldsymbol{x}_{s_{2,k}})}{K_2} - f_d(\boldsymbol{x})}_{\text{term 2}} \right) \right|}{2\sigma_d(\boldsymbol{x})^2}, \tag{51}$$

with $s_{1,k},\ s_{2,k} \sim \mathrm{P}(S)$, and $d \sim \mathrm{U}(1, D)$. We denote $\hat{\phi}_{j,d} = \frac{\sum_{k=1}^{K_1} f(\boldsymbol{x}_{s_{j,k} \cup d}) - f(\boldsymbol{x}_{s_{j,k}})}{K_j}$, and write:

$$\hat{D}_{SHAP} = D \frac{\left| \left( \underbrace{\hat{\phi}_{1,d} - f_d(\boldsymbol{x})}_{\text{term 1}} \right) \left( \underbrace{\hat{\phi}_{2,d} - f_d(\boldsymbol{x})}_{\text{term 2}} \right) \right|}{2\sigma_d(\boldsymbol{x})^2}. \tag{52}$$

When computing stochastic estimates $\hat{D}_{SHAP}$, there are two possible cases that can occur: either both terms in the numerator have the same sign, or they have different signs. We denote these as two disjoint events:

$$\begin{cases} \mathcal{A} := sign(\text{term 1}) = sign(\text{term 2}) \,, \\ \mathcal{B} := sign(\text{term 1}) \neq sign(\text{term 2}) \,. \end{cases} \tag{53}$$

To compute the probabilities of such events, we first define intermediate probabilities as determined by the stochastic $\hat{\phi}_{j,d}$ terms

$$\mathrm{P}\left(\hat{\phi}_{1,d} > f_d(\boldsymbol{x})\right) = \mathrm{P}\left(\hat{\phi}_{2,d} > f_d(\boldsymbol{x})\right) = p \,, \tag{54}$$

$$\mathrm{P}\left(\hat{\phi}_{1,d} < f_d(\boldsymbol{x})\right) = \mathrm{P}\left(\hat{\phi}_{2,d} < f_d(\boldsymbol{x})\right) = 1 - p \,, \tag{55}$$

to conclude that

$$\mathrm{P}(\mathcal{A}) = \mathrm{P}\left(\hat{\phi}_{1,d} > f_d(\boldsymbol{x})\right)\mathrm{P}\left(\hat{\phi}_{2,d} > f_d(\boldsymbol{x})\right) + \mathrm{P}\left(\hat{\phi}_{1,d} < f_d(\boldsymbol{x})\right)\mathrm{P}\left(\hat{\phi}_{2,d} < f_d(\boldsymbol{x})\right) \tag{56}$$

$$= p^2 + (1 - p)^2 \,, \tag{57}$$

$$\mathrm{P}(\mathcal{B}) = \mathrm{P}\left(\hat{\phi}_{1,d} < f_d(\boldsymbol{x})\right)\mathrm{P}\left(\hat{\phi}_{2,d} > f_d(\boldsymbol{x})\right) + \mathrm{P}\left(\hat{\phi}_{1,d} > f_d(\boldsymbol{x})\right)\mathrm{P}\left(\hat{\phi}_{2,d} < f_d(\boldsymbol{x})\right) \tag{58}$$

$$= 2p(1 - p) \,. \tag{59}$$

We investigate each of these two events separately below.

**Event $\mathcal{A}$: Both terms have the same sign.**

$$\mathbb{E}_{\mathrm{P}(s_1)\mathrm{P}(s_2)\mathrm{U}(1,D)}\left\{\hat{D}_{SHAP}|\mathcal{A}\right\} = \mathbb{E}_{\mathrm{P}(s_1)\mathrm{P}(s_2)\mathrm{U}(1,D)}\left\{D\frac{\left(\hat{\phi}_{1,d} - f_d(\boldsymbol{x})\right)\left(\hat{\phi}_{2,d} - f_d(\boldsymbol{x})\right)}{2\sigma_d(\boldsymbol{x})^2}\right\} \tag{60}$$

$$= D\mathbb{E}_{\mathrm{P}(s_1)\mathrm{P}(s_2)\mathrm{U}(1,D)}\left\{\frac{\hat{\phi}_{1,d}\hat{\phi}_{2,d} - f_d(\boldsymbol{x})\hat{\phi}_{1,d} - f_d(\boldsymbol{x})\hat{\phi}_{2,d} + f_d^2(\boldsymbol{x})}{2\sigma_d(\boldsymbol{x})^2}\right\} \tag{61}$$

$$= D\sum_{d=1}^{D}\frac{1}{D}\frac{\mathbb{E}_{\mathrm{P}(s_1)}\left\{\hat{\phi}_{1,d}\right\}\mathbb{E}_{\mathrm{P}(s_2)}\left\{\hat{\phi}_{2,d}\right\} - f_d(\boldsymbol{x})\mathbb{E}_{\mathrm{P}(s_1)}\left\{\hat{\phi}_{1,d}\right\} - f_d(\boldsymbol{x})\mathbb{E}_{\mathrm{P}(s_2)}\left\{\hat{\phi}_{2,d}\right\} + f_d^2(\boldsymbol{x})}{2\sigma_d(\boldsymbol{x})^2} \tag{62}$$

Because $\phi_d = \mathbb{E}_{\mathrm{P}(s_1)}\left\{\hat{\phi}_{1,d}\right\} = \mathbb{E}_{\mathrm{P}(s_2)}\left\{\hat{\phi}_{2,d}\right\} = \sum_{S\subseteq[D]\setminus\{d\}}\mathrm{P}(S)\left(f(\boldsymbol{x}_{S\cup d}) - f(\boldsymbol{x}_s)\right)$

$$= D\sum_{d=1}^{D}\frac{1}{D}\frac{\phi_d^2 - 2f_d(\boldsymbol{x})\phi_d + f_d^2(\boldsymbol{x})}{2\sigma_d(\boldsymbol{x})^2} \tag{63}$$

$$= D\sum_{d=1}^{D}\frac{1}{D}\frac{(\phi_d - f_d(\boldsymbol{x}))^2}{2\sigma_d(\boldsymbol{x})^2} \tag{64}$$

$$= D_{\mathrm{SHAP}} \tag{65}$$

**Event $\mathcal{B}$: The terms have opposite signs.**

$$\mathbb{E}_{\mathrm{P}(s_1)\mathrm{P}(s_2)\mathrm{U}(1,D)}\left\{\hat{D}_{SHAP}|\mathcal{B}\right\} = \mathbb{E}_{\mathrm{P}(s_1)\mathrm{P}(s_2)\mathrm{U}(1,D)}\left\{D\frac{\left(f_d(\boldsymbol{x}) - \hat{\phi}_{1,d}\right)\left(\hat{\phi}_{2,d} - f_d(\boldsymbol{x})\right)}{2\sigma_d(\boldsymbol{x})^2}\right\} \tag{66}$$

$$= D\mathbb{E}_{\mathrm{P}(s_1)\mathrm{P}(s_2)\mathrm{U}(1,D)}\left\{\frac{-\hat{\phi}_{1,d}\hat{\phi}_{2,d} + f_d(\boldsymbol{x})\hat{\phi}_{1,d} + f_d(\boldsymbol{x})\hat{\phi}_{2,d} - f_d^2(\boldsymbol{x})}{2\sigma_d(\boldsymbol{x})^2}\right\} \tag{67}$$

$$= D\sum_{d=1}^{D}\frac{1}{D}\frac{-\mathbb{E}_{\mathrm{P}(s_1)}\left\{\hat{\phi}_{1,d}\right\}\mathbb{E}_{\mathrm{P}(s_2)}\left\{\hat{\phi}_{2,d}\right\} + f_d(\boldsymbol{x})\mathbb{E}_{\mathrm{P}(s_1)}\left\{\hat{\phi}_{1,d}\right\} + f_d(\boldsymbol{x})\mathbb{E}_{\mathrm{P}(s_2)}\left\{\hat{\phi}_{2,d}\right\} - f_d^2(\boldsymbol{x})}{2\sigma_d(\boldsymbol{x})^2} \tag{68}$$

Because $\phi_d = \mathbb{E}_{\mathrm{P}(s_1)}\left\{\hat{\phi}_{1,d}\right\} = \mathbb{E}_{\mathrm{P}(s_2)}\left\{\hat{\phi}_{2,d}\right\} = \sum_{S\subseteq[D]\setminus\{d\}}\mathrm{P}(S)\left(f(\boldsymbol{x}_{S\cup d}) - f(\boldsymbol{x}_s)\right)$

$$= D\sum_{d=1}^{D}\frac{1}{D}\frac{-\phi_d^2 + 2f_d(\boldsymbol{x})\phi_d - f_d^2(\boldsymbol{x})}{2\sigma_d(\boldsymbol{x})^2} \tag{69}$$

$$= \sum_{d=1}^{D}\frac{1}{D}\frac{-(\phi_d - f_d(\boldsymbol{x}))^2}{2\sigma_d(\boldsymbol{x})^2} \tag{70}$$

$$= -D_{\mathrm{SHAP}} \tag{71}$$

**The expected value of $\hat{D}_{SHAP}$.** We compute the overall expected value of $\hat{D}_{SHAP}$, using the law of total expectation:

$$\mathbb{E}\left\{\hat{D}_{SHAP}\right\} = \mathbb{E}_{P(\mathcal{A},\mathcal{B})}\left\{\mathbb{E}_{P(s_1)P(s_2)U(1,D)}\left\{\hat{D}_{SHAP}|\mathcal{A},\mathcal{B}\right\}\right\} \tag{72}$$

$$= P(\mathcal{A})\left\{\mathbb{E}_{P(s_1)P(s_2)U(1,D)}\left\{\hat{D}_{SHAP}|\mathcal{A}\right\}\right\} + P(\mathcal{B})\left\{\mathbb{E}_{P(s_1)P(s_2)U(1,D)}\left\{\hat{D}_{SHAP}|\mathcal{B}\right\}\right\} \tag{73}$$

$$= P(\mathcal{A})D_{SHAP} + P(\mathcal{B})(-D_{SHAP}) \tag{74}$$

$$= D_{SHAP}(P(\mathcal{A}) - P(\mathcal{B})) \tag{75}$$

$$= D_{SHAP}\left(p^2 + (1-p)^2 - 2p(1-p)\right) \tag{76}$$

$$= D_{SHAP}\left(p^2 + 1 - 2p + p^2 - 2p + 2p^2\right) \tag{77}$$

$$= D_{SHAP}\left(4p^2 - 4p + 1\right) \tag{78}$$

First, we note that $\left(4p^2 - 4p + 1\right) \geq 0$. Hence, $\hat{D}_{SHAP}$ is a biased estimator in general, where the bias is strictly positive: $\left(4p^2 - 4p + 1\right) \geq 0$. Therefore, the gradients of $D_{SHAP}$ and $\hat{D}_{SHAP}$ are always proportional to each other, and point to the same direction —a key property for the optimization of the proposed loss.

Second, $\mathbb{E}\left\{\hat{D}_{SHAP}\right\} = D_{SHAP}$ when $p = 0$. Hence, $\mathbb{E}\left\{\hat{D}_{SHAP}\right\} = D_{SHAP} = 0$ only when $f_d(\boldsymbol{x}) = \phi_d$. Namely, $\hat{D}_{SHAP} = D_{SHAP} = 0$, only when the model's Shapley prior is centered exactly at the function values.

## E  The MENN architecture: A Masking Operation Example

In this section, we demonstrate an illustrative example masking matrices and operations. For simplicity, we use a four layer MENN, with hidden dimensions $[10, 10, 10]$, and input/output dimensions 3 and 6, respectively. This set-up corresponds to two-dimensional embeddings per input feature. We begin by defining our initial masking matrix $\mathbf{M}_1$:

$$\mathbf{M}_1 = \begin{bmatrix} 1 & 1 & 1 & 0 & 0 & 0 & 0 & 0 & 0 & 0 \\ 0 & 0 & 0 & 1 & 1 & 1 & 0 & 0 & 0 & 0 \\ 0 & 0 & 0 & 0 & 0 & 0 & 1 & 1 & 1 & 1 \end{bmatrix}$$

Moving deeper in the network, we define intermediate masking matrices $\mathbf{M}_k$ by repeating the rows of $\mathbf{M}_{1:k-1}$ as described in Section 3.4.1:

$$\mathbf{M}_2 = \begin{bmatrix} 1 & 1 & 1 & 0 & 0 & 0 & 0 & 0 & 0 & 0 \\ 1 & 1 & 1 & 0 & 0 & 0 & 0 & 0 & 0 & 0 \\ 1 & 1 & 1 & 0 & 0 & 0 & 0 & 0 & 0 & 0 \\ 0 & 0 & 0 & 1 & 1 & 1 & 0 & 0 & 0 & 0 \\ 0 & 0 & 0 & 1 & 1 & 1 & 0 & 0 & 0 & 0 \\ 0 & 0 & 0 & 1 & 1 & 1 & 0 & 0 & 0 & 0 \\ 0 & 0 & 0 & 0 & 0 & 0 & 1 & 1 & 1 & 1 \\ 0 & 0 & 0 & 0 & 0 & 0 & 1 & 1 & 1 & 1 \\ 0 & 0 & 0 & 0 & 0 & 0 & 1 & 1 & 1 & 1 \\ 0 & 0 & 0 & 0 & 0 & 0 & 1 & 1 & 1 & 1 \end{bmatrix}$$

$$\mathbf{M}_3 = \begin{bmatrix} 1 & 1 & 1 & 0 & 0 & 0 & 0 & 0 & 0 & 0 \\ 1 & 1 & 1 & 0 & 0 & 0 & 0 & 0 & 0 & 0 \\ 1 & 1 & 1 & 0 & 0 & 0 & 0 & 0 & 0 & 0 \\ 0 & 0 & 0 & 1 & 1 & 1 & 0 & 0 & 0 & 0 \\ 0 & 0 & 0 & 1 & 1 & 1 & 0 & 0 & 0 & 0 \\ 0 & 0 & 0 & 1 & 1 & 1 & 0 & 0 & 0 & 0 \\ 0 & 0 & 0 & 0 & 0 & 0 & 1 & 1 & 1 & 1 \\ 0 & 0 & 0 & 0 & 0 & 0 & 1 & 1 & 1 & 1 \\ 0 & 0 & 0 & 0 & 0 & 0 & 1 & 1 & 1 & 1 \\ 0 & 0 & 0 & 0 & 0 & 0 & 1 & 1 & 1 & 1 \end{bmatrix}$$

and

$$\mathbf{M}_4 = \begin{bmatrix} 1 & 1 & 0 & 0 & 0 & 0 \\ 1 & 1 & 0 & 0 & 0 & 0 \\ 1 & 1 & 0 & 0 & 0 & 0 \\ 0 & 0 & 1 & 1 & 0 & 0 \\ 0 & 0 & 1 & 1 & 0 & 0 \\ 0 & 0 & 1 & 1 & 0 & 0 \\ 0 & 0 & 0 & 0 & 1 & 1 \\ 0 & 0 & 0 & 0 & 1 & 1 \\ 0 & 0 & 0 & 0 & 1 & 1 \\ 0 & 0 & 0 & 0 & 1 & 1 \end{bmatrix}$$

Observe that $\mathbf{M}_4$ distributes 2 embedding dimensions per feature.

Now we compute the following matrix multiplication with non-zero elements of $\mathbf{M}_{1:k}$ denoting the information flow until layer $k$ (Germain et al., 2015):

$$\mathbf{M}_{1:2} = \begin{bmatrix} 3 & 3 & 3 & 0 & 0 & 0 & 0 & 0 & 0 & 0 \\ 0 & 0 & 0 & 3 & 3 & 3 & 0 & 0 & 0 & 0 \\ 0 & 0 & 0 & 0 & 0 & 0 & 4 & 4 & 4 & 4 \end{bmatrix}$$

Notice that the columns of $\mathbf{M}_3$ have been constructed by repeating the rows of binarized $\mathbf{M}_{1:2}$ (compare) as described in the main manuscript (i.e., rows of $\mathbf{M}_3$ are constructed by repeating the rows of $\mathbf{M}'_{1:2}$, where each element in $\mathbf{M}'_{1:2}$ is defined by $\mathbf{M}'_{1:2}(d,l) = \mathbf{1}_{\mathbf{M}_{1:2}>0}(d,l)$). We now compute $\mathbf{M}_{1:3}$:

$$\mathbf{M}_{1:3} = \begin{bmatrix} 9 & 9 & 9 & 0 & 0 & 0 & 0 & 0 & 0 & 0 \\ 0 & 0 & 0 & 9 & 9 & 9 & 0 & 0 & 0 & 0 \\ 0 & 0 & 0 & 0 & 0 & 0 & 16 & 16 & 16 & 16 \end{bmatrix}$$

We finally compute

$$\mathbf{M}_{1:4} = \begin{bmatrix} 27 & 27 & 0 & 0 & 0 & 0 \\ 0 & 0 & 27 & 27 & 0 & 0 \\ 0 & 0 & 0 & 0 & 64 & 64 \end{bmatrix} = \begin{bmatrix} \gamma_1 & \gamma_1 & 0 & 0 & 0 & 0 \\ 0 & 0 & \gamma_2 & \gamma_2 & 0 & 0 \\ 0 & 0 & 0 & 0 & \gamma_3 & \gamma_3 \end{bmatrix}$$

Notice that $\mathbf{M}_{1:4}$ allows for creating two dimensional embeddings as it permits information flow to two disjoint output neurons for each input feature: i.e., outputs one and two only depend on $x_1$, outputs three and four only depend on $x_2$, and output three only depends on $x_3$.

Observe how $\gamma_i$ values depend on how the 1s are distributed in the masking matrices; an equal distribution would result in more uniform $\gamma_i$ values. In practice, instead of using a single $nint(.)$ function, we choose to round up or down, randomly.

## F  PSI evaluation: Experimental set-up and additional details

### F.1  Simulated Data Generating Processes

We describe below the data generating processes (DGPs) of the simulated datasets, where access to ground-truth feature attribution information is available. We use non-trivial functions for the heteroscedastic noise, and impose complex feature-interactions. We generate 8000 examples for each synthetic dataset.

**Synthetic1.** DGP is as described in Section 4.1: $(i)$ Draw $x_1, x_2, x_3 \sim \mathrm{U}(-4,4)$, $(ii)$ Draw $f_1 \sim \mathcal{N}\left(2 + \exp\left\{-x_1^2\right\} - \frac{\sqrt{\pi}}{8} erf\{4\}, 0.6\cos{(0.03x_1)}^{800}\right)$, $f_2 \sim \mathcal{N}\left(1 + \sin{(-x_2^2)} + \frac{\sqrt{2\pi}}{8}S(4\sqrt{\frac{2}{\pi}}), 0.2|x_2|\right)$, and $f_3 = 3\cos(3x_3) + 4\sin(5x_3)$, $(iii)$ Observe $y = f_1 + f_2 + f_3$. Here, $S(.)$ is the Fresnel integral.

**Synthetic2.** $(i)$ Draw $x_1, x_2, x_3 \sim \mathrm{U}(-4,4)$, $(ii)$ Calculate $f_1 = \exp\left\{-x_1^2\right\}x_1$, $f_2 = 0.5x_2\sin(x_2)$, $f_3 = \cos(3x_3)\sin(x_3)$, $(iii)$ Observe $y = f_1 + f_2 + f_3$.

**Synthetic3.** ($i$) Draw $x_1, x_2, x_3 \sim$ U$(-4, 4)$, ($ii$) Calculate $f_1 = 4\sin(x_1) + 2\sin(2x_1)$, $f_2 = 3\cos(3x_2)\sin(5x_2)$, $f_3 = \cos(2x_3) + x_3^2/7$, $f_{12} = \exp\{-(x_1 + x_2)^2\}$, $f_{13} = (x_1 - 3)x_3\sin(x_1)\cos(x_3)/2$, $f_{23} = x_2 x_3/2$, ($iii$) Observe random $y \sim \mathcal{N}(f_1 + f_2 + f_3 + f_{12} + f_{13} + f_{23}, 0.01)$.

**Synthetic4.** ($i$) Draw $x_1, x_2, x_3 \sim$ U$(-4, 4)$, ($ii$) Calculate $f_1 = \exp\{-1/x_1^2\} + \sin(100/x_1)$), $f_2 = \exp\{-|\cos(|x_2|) + 1/2\sin(2x_2)|\} + x_2/4$, $f_3 = \tanh(x_3^2)$, $f_{12} = \sin(x_1^2 + x_2^2)/2$, ($iii$) Observe $y = f_1 + f_2 + f_3 + f_{12}$.

**Synthetic5.** ($i$) Draw $x_1, x_2, x_3 \sim$ U$(-4, 4)$, ($ii$) Calculate $f_1 = |x_1|/10 + x_1^2/10 + \sin(x_1)$, $f_2 = \cos(5x_2) + \sin(2x_2) + x_2$, $f_3 = \exp\{-x_3^{100}\}$, $f_{12} = 5(x_1^{10} + x_2^{10})^{1/10}/2$, $f_{23} = 5|\sin(x_3 x_2)\cos(x_3 x_2)|/2$, ($iii$) Observe random $y \sim \mathcal{N}(f_1 + f_2 + f_3 + f_{12}, 0.01)$.

The observed data consists of only $\{\boldsymbol{x}, y\}_{i=1}^N$ pairs for all simulated datasets.

## F.2 Real-world Datasets

**Regression Datasets.** We describe here the datasets used in regression tasks:

1. **Parkinsons telemonitoring (PKSN):** The dataset features voice data from 42 Parkinson's patients in the early stages, captured over six months using a telemonitoring device for distant symptom monitoring. These autonomous recordings, taken at their homes, total 5875 data points. The task is to predict Clinician's motor UPDRS score (Tsanas et al., 2009).

2. **Medical expenses (MED):** 1,338 patients from United States (Lantz, 2019). The task is to predict medical expenses. Here, the unit of measurement is United States Dolars (USD).

3. **Bike sharing (BIKE):** The dataset of 17,389 entries aims to predict total bike rentals, covering both casual and registered users (Fanaee-T, 2013).

4. **Greenhouse gas observing network (GAS):** The dataset features time series of greenhouse gas concentrations across 2,921 grid cells in California, generated using the WRF-Chem simulation model. The goal is to predict the green house gas concentration (Lucas, 2015).

**Classification Datasets.** These are the datasets used in classification tasks:

1. **Heloc (FICO):** 9,861 credit applications (FICO, 2018). The task is to classify risk performance. The good risk performance is represented by 1.

2. **Spambase (SPAM):** The classification task involves 4601 instances aimed at determining whether a given email is classified as spam or not (Hopkins et al., 1999). Spam emails are represented by 1.

3. **Intensive care unit (ICU):** 15,830 intensive care unit (ICU) cases from Argentina, Australia, New Zealand, Sri Lanka, Brazil, and United States (Meredith et al., 2020). The task is to classify ICU mortality. The ICU mortality is represented by 1.

4. **Census income (CENS):** Demographic (such as age, education etc.) information of 48,842 people. The task is to predict if the person earns more than 50,000 USD a year (Kohavi, 1996). Income of more than 50,000 USD a year is represented by 1.

## F.3 Baseline Models

1. **Bayesian Linear/Logistic Regression (LIN):** The simplest, yet explainable model. Linear models quantify the feature importance and feature importance uncertainty through the model coefficients. We use Bayesian linear regression for regression and Bayesian logistic regression for classification tasks. We use the implementation by Pedregosa et al. (2011).

2. **Explainable Boosting Machines (EBM):** The state-of-the-art explainable additive model which uses ensemble shallow trees with boosting to model each component (Lou et al., 2013; Caruana et al., 2015). We use the implementation by Nori et al. (2019).

3. **Random Forest (RF):** Another ensemble learning algorithm that works by building multiple trees independently using bagging, and averaging the predictions of each individual tree. We use the implementation by Pedregosa et al. (2011).

4. **Light Gradient Boosting Machines (LGBM):** LGBM uses a leaf-wise growth strategy, prioritizing splits that result in the largest decrease in loss, whereas most traditional tree-based algorithms grow trees level-wise. While leaf-wise growth can achieve better accuracy, it might also lead to overfitting, especially on smaller datasets. Thus, careful hyperparameter tuning, including regularization, is essential when using LGBM (Ke et al., 2017).

5. **Gradient Boosted Trees (XGB):** A well-known ensemble learning algorithm that combines several week learners. In particular, each weak learner is trained to improve the ensemble performance, one at a time. We use the implementation by Chen & Guestrin (2016).

6. **Deep Neural Network (DNN):** Universal function approximators, where DNNs relate input and output through non-linear mappings.

### F.4 Baseline Hyperparameters

We sample 300 hyperparameters, train each model on test set and evaluate on the validation set to find the optimum parameters for learning a train-test split. We then test the models on the remaining test fold. We do this 5 times. We use NVIDIA RTX 2080 graphics card for training neural models.

The hyperparameter search space of all models are as follows:

LIN

- Regressor
  param_grid = {
  "C": [5e-2, 1e-1, 5e-1, 1],
  "max_iter":[5000]
  }

- Classsifier
  param_grid = {
  "C": [5e-2, 1e-1, 5e-1, 1],
  "max_iter":[5000]
  }

EBM

- Regressor
  param_grid = {
  "outer_bags": [8, 25, 75, 100],
  "inner_bags":[0, 1, 2, 5, 10]
  }

- Classsifier
  param_grid = {
  "outer_bags": [8, 25, 75, 100],
  "inner_bags":[0, 1, 2, 5, 10]
  }

RF

- Regressor
  param_grid = {
  "ccp_theta": [0.0, 1e-1, 1e-2],
  "max_depth": [None, 4, 8, 16, 40, 100],
  "min_samples_leaf": [1, 3, 5, 10],
  "min_samples_split": [2, 4, 6, 12],
  "n_estimators": [100, 200, 600, 800]
  }

- Classsifier
  param_grid = {
  "ccp_theta": [0.0, 1e-1, 1e-2],
  "max_depth": [None, 4, 8, 16, 40, 100],
  "min_samples_leaf": [1, 3, 5, 10],
  "min_samples_split": [2, 4, 6, 12],
  "n_estimators": [100, 200, 600, 800]
  }

LGBM

- Regressor
  param_grid = {
  "num_leaves": [31, 50, 70, 100],
  "max_depth": [-1, 5, 7, 10],
  "learning_rate": [0.001, 0.01, 0.05, 0.1],
  "n_estimators": [100, 200, 500],
  "min_split_gain": [0.0, 0.1, 0.5],
  "min_child_weight": [1e-3, 1e-2, 1e-1, 1],
  "min_child_samples": [20, 30],
  "subsample": [0.8, 0.9, 1.0],
  "colsample_bytree": [0.7, 0.8, 0.9, 1.0],
  "reg_theta": [0, 1, 2],
  "reg_lambda": [0, 1, 2],
  "boosting_type": ['gbdt', 'dart'],
  }

- Classsifier
  param_grid = {
  "num_leaves": [31, 50, 70, 100],
  "max_depth": [-1, 5, 7, 10],
  "learning_rate": [0.001, 0.01, 0.05, 0.1],
  "n_estimators": [100, 200, 500],
  "min_split_gain": [0.0, 0.1, 0.5],
  "min_child_weight": [1e-3, 1e-2, 1e-1, 1],
  "min_child_samples": [20, 30],
  "subsample": [0.8, 0.9, 1.0],
  "colsample_bytree": [0.7, 0.8, 0.9, 1.0],
  "reg_theta": [0, 1, 2],
  "reg_lambda": [0, 1, 2],
  "boosting_type": ['gbdt', 'dart'],
  }

XGB

- Regressor param_grid = {
  "learning_rate" : [0.05, 0.10, 0.15, 0.30],
  "max_depth" : [3, 5, 8, 15],
  "min_child_weight" : [1, 3, 7],
  "theta" : [0.0, 0.1, 0.3],
  "colsample_bytree" : [0.3, 0.4, 0.5],
  'ccp_theta': [0.0,1e-3,1e-2],
  "min_impurity_decrease": [0, 1e-1]
  }

- Classsifier
  param_grid = {
  "learning_rate" : [0.05, 0.10, 0.15, 0.30],
  "max_depth" : [3, 5, 8, 15],
  "min_child_weight" : [1, 3, 7],
  "theta" : [0.0, 0.1, 0.3],
  "colsample_bytree" : [0.3, 0.4, 0.5],
  'ccp_theta': [0.0,1e-3,1e-2],
  "min_impurity_decrease": [0, 1e-1]
  }

DNN

- Regressor
  param_grid = { "batch_size": [1024, 512],
  "lr":[5e-4, 1e-3, 2e-3],
  "act": ['relu', 'snake', 'elu'],
  "norm":[None, 'layer', 'batch'],
  "n_layers":[2, 3, 4, 5],
  "d_hid":[25, 50, 75, 100, 200],
  "weight_decay": [0, 1e-10, 1e-8, 1e-6],
  "dropout": [0, 0.2, 0.4, 0.5],
  }

- Classsifier
  param_grid = { "batch_size": [1024, 512],
  "lr":[5e-4, 1e-3, 2e-3],
  "act": ['relu', 'snake', 'elu'],
  "norm":[None, 'layer', 'batch'],
  "n_layers":[2, 3, 4, 5],
  "d_hid":[25, 50, 75, 100, 200],
  "weight_decay": [0, 1e-10, 1e-8, 1e-6],
  "dropout": [0, 0.2, 0.4, 0.5],
  }

PSI

- Regressor
  param_grid = { "batch_size": [1024, 512, 256],
  "lr":[2e-3, 1e-3, 5e-4],
  "beta":[10, 1, 0.1, 0.01, 0.001],
  "act": ['relu', 'snake', 'elu'],
  "arch":[FFNN, MENN'],
  "norm":[None, 'layer'],
  "n_layers":[5, 4, 3, 2],
  "d_hid":[300, 200, 150, 100],
  "weight_decay": [1e-6, 1e-7, 1e-8, 0],
  "dropout": [0],
  "p_missing":[1/2, 2/3, 'shapley'],
  }

- Classsifier
  param_grid = { "batch_size": [1024, 512, 256],
  "lr":[2e-3, 1e-3, 5e-4],
  "beta":[10, 1, 0.1, 0.01, 0.001],
  "act": ['relu', 'snake', 'elu'],
  "arch":[FFNN, MENN'],
  "norm":[None, 'layer'],
  "n_layers":[5, 4, 3, 2],
  "d_hid":[300, 200, 150, 100],
  "weight_decay": [1e-6, 1e-7, 1e-8, 0],
  "dropout": [0],
  "p_missing":[1/2, 2/3, 'shapley'],
  }

"p_missing" refers to the probability of introducing a feature to the network during training (Jethani et al., 2021).

## F.5 Predictive performance results

| Data | Interpretable | | | Black-box | | | |
|---|---|---|---|---|---|---|---|
| | PSI | LIN | EBM | RF | LGBM | XGB | DNN |
| PKSN | $0.026^{\pm0.002}$ | $0.867^{\pm0.009}$ | $0.195^{\pm0.002}$ | $0.040^{\pm0.006}$ | $0.072^{\pm0.009}$ | $0.063^{\pm0.002}$ | $0.111^{\ \pm0.005}$ |
| MED | $0.447^{\pm0.018}$ | $0.608^{\pm0.015}$ | $0.447^{\pm0.021}$ | $0.448^{\pm0.018}$ | $0.447^{\pm0.020}$ | $0.454^{\pm0.016}$ | $0.467^{\pm0.005}$ |
| BIKE | $0.019^{\pm0.001}$ | $0.518^{\pm0.004}$ | $0.038^{\pm0.001}$ | $0.008^{\pm0.001}$ | $0.017^{\pm0.002}$ | $0.013^{\pm0.001}$ | $0.019^{\pm0.001}$ |
| GAS | $0.269^{\pm0.045}$ | $60.93^{\pm60.060}$ | $0.153^{\pm0.016}$ | $0.151^{\pm0.015}$ | $0.152^{\pm0.012}$ | $0.150^{\pm0.021}$ | $0.256^{\pm0.019}$ |
| FICO | $0.771^{\pm0.004}$ | $0.766^{\pm0.004}$ | $0.771^{\pm0.004}$ | $0.770^{\pm0.003}$ | $0.773^{\pm0.005}$ | $0.773^{\pm0.004}$ | $0.771^{\pm0.007}$ |
| SPAM | $0.977^{\pm0.006}$ | $0.946^{\pm0.006}$ | $0.977^{\pm0.003}$ | $0.983^{\pm0.002}$ | $0.982^{\pm0.002}$ | $0.981^{\pm0.003}$ | $0.967^{\pm0.005}$ |
| ICU | $0.872^{\pm0.004}$ | $0.851^{\pm0.004}$ | $0.872^{\pm0.004}$ | $0.869^{\pm0.004}$ | $0.874^{\pm0.003}$ | $0.874^{\pm0.003}$ | $0.860^{\pm0.004}$ |
| CENS | $0.803^{\pm0.003}$ | $0.768^{\pm0.003}$ | $0.825^{\pm0.003}$ | $0.800^{\pm0.002}$ | $0.830^{\pm0.003}$ | $0.831^{\pm0.004}$ | $0.788^{\pm0.003}$ |

Table 3: PSI and baseline comparison across real-world datasets with standard deviation results.

| Net | $D\beta/2$ | synth1 | synth2 | synth3 | synth4 | synth5 |
|---|---|---|---|---|---|---|
| MENN | 0.001 | $0.605^{\pm0.011}$ | $0.004^{\pm0.000}$ | $0.125^{\pm0.006}$ | $0.669^{\pm0.017}$ | $0.419^{\pm0.003}$ |
| | 0.01 | $0.611^{\pm0.014}$ | $0.004^{\pm0.000}$ | $0.120^{\pm0.002}$ | $0.669^{\pm0.018}$ | $0.416^{\pm0.010}$ |
| | 0.1 | $0.595^{\pm0.011}$ | $0.004^{\pm0.000}$ | $0.221^{\pm0.006}$ | $0.673^{\pm0.016}$ | $0.417^{\pm0.003}$ |
| | 1 | $0.582^{\pm0.009}$ | $0.004^{\pm0.000}$ | $1.118^{\pm0.027}$ | $0.579^{\pm0.029}$ | $0.446^{\pm0.005}$ |
| FFNN | 0.001 | $0.642^{\pm0.012}$ | $0.071^{\pm0.063}$ | $0.368^{\pm0.155}$ | $0.689^{\pm0.022}$ | $0.433^{\pm0.008}$ |
| | 0.01 | $0.640^{\pm0.009}$ | $0.128^{\pm0.074}$ | $0.234^{\pm0.078}$ | $0.677^{\pm0.018}$ | $0.435^{\pm0.007}$ |
| | 0.1 | $0.624^{\pm0.011}$ | $0.103^{\pm0.059}$ | $0.174^{\pm0.006}$ | $0.674^{\pm0.014}$ | $0.431^{\pm0.007}$ |
| | 1 | $0.599^{\pm0.008}$ | $0.186^{\pm0.110}$ | $0.897^{\pm0.024}$ | $0.660^{\pm0.017}$ | $0.477^{\pm0.005}$ |

Table 4: Comparison of FFNN and MENN architectures with fold-wise standard deviation results.

## F.6 Jeffreys Divergence

The metric that we are interested in is:

$$\frac{1}{D} \sum_{j\in[D]} \sum_{s\in\mathcal{A}\setminus\{j\}} \mathrm{P}(s) \times \text{J-Divergence}(\text{Model Marginals}, \text{Empirical Marginals}), \qquad (79)$$

which calculates the distance between marginals for every possible coalition that can be computed for any given feature $j$, weighted according to its Shapley value weights. This allows for arriving at a single metric calculated by weighting the distance between ground truth marginals and model marginals by their occurrence in Shapley value calculations.

Precisely, we start by setting $dist = 0$ and, for every feature $j \in [D]$ and $s \in s \in [D] \setminus \{j\}$, we follow the steps below:

1. **Empirical sampling:** Sample $[y, \boldsymbol{x}] \sim \mathbb{D}$ from the empirical data joint distribution.

2. **Empirical marginals:** Remove the columns $k \notin s$ from $[y, \boldsymbol{x}]$ to obtain empirical marginals $[y, \boldsymbol{x}_s]$.

3. **Sample from model marginals:** Sample $\hat{y} \sim \mathrm{P}(y \mid \boldsymbol{x}_s)$ from PSI using MENN/FFNN, and denote $[\hat{y}, \boldsymbol{x}_s]$ as PSI marginal samples.

4. **Measure distance:** Compute $dist = dist + \frac{1}{D}\mathrm{P}(s) \times \text{J-Divergence}(p([y, \boldsymbol{x}_s]), p([\hat{y}, \boldsymbol{x}_s]))$.

**Computing the J-Divergence** $(p([y, \boldsymbol{x}_s]), p([\hat{y}, \boldsymbol{x}_s]))$. We estimate $p([y, \boldsymbol{x}_s])$ and $p([\hat{y}, \boldsymbol{x}_s])$ by discretizing the empirical distribution defined by the available model and data samples, and fit a histogram of different

bin sizes to both. For such discretized distributions, the J-Divergence has a closed-form solution (since the KL-terms can be computed analytically), that can be computed:

$$\frac{1}{2}\left(D_{\mathrm{KL}}\left(q\|p\right)+D_{\mathrm{KL}}\left(p\|q\right)\right),\ \text{where}\ D_{\mathrm{KL}}\left(q\|p\right)=\sum q\log p,\tag{80}$$

where the sum is carried out over the histogram bins. $p$ and $q$ here represent each bin's probability.

We apply this same procedure for various, random bin sizes: we sample 20 bin sizes from $U(10,200)$. We then take the average over divergences over all bin-size histograms, resulting in a single number estimator for the J-Divergence metric of interest.

The error bars shown in Figure 7 were generated by applying the same procedure described above to 5 different models trained of different train-test splits (i.e., 5 fold cross-validation).

## G  Empirical evidence of convergence as in Remark 3.3

We provide empirical evidence of Remark 3.3 in Figure 8: the empirical estimates $\mathbb{E}_{\boldsymbol{x}\sim\mathbb{D}}\left\{f_d(\boldsymbol{x})\right\}$ converge to zero as the model learns for both MENN and FFNN architectures —an effect most prominent for $\beta=1$.

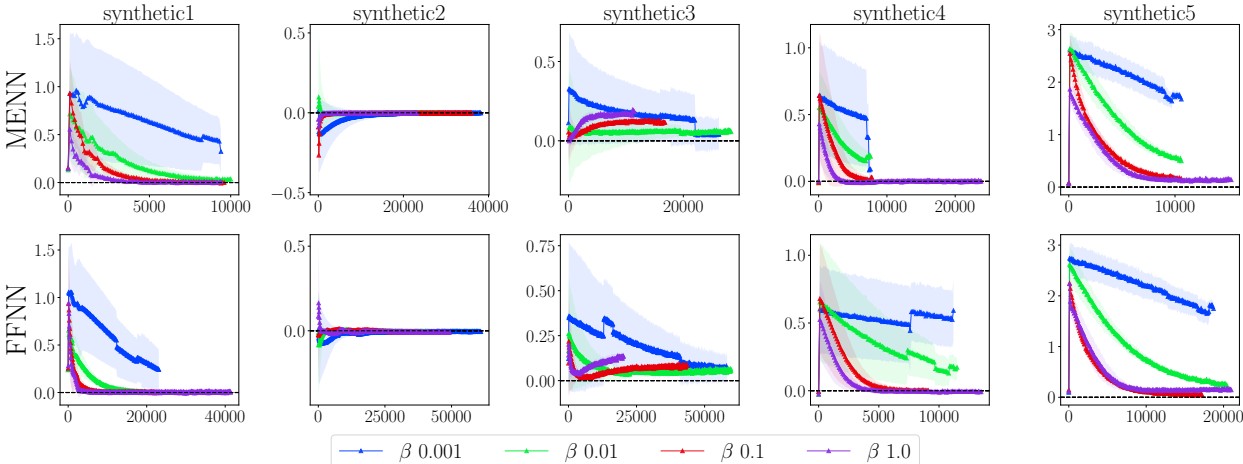

Figure 8: Average of the $f_d$ over the empirical dataset, $\mathbb{E}_{\boldsymbol{x}\sim\mathbb{D}}\left\{f_d(\boldsymbol{x})\right\}$, w.r.t. training epochs. Shaded regions are the error values from different cross-validation folds.

