# OpenReview forum: "Probabilistic Shapley Value Modeling and Inference"
_TMLR — Withdrawn by Authors_

### Review · Reviewer_56Cc · 2025-12-19

**Summary Of Contributions:**

The paper proposes a new Shapley-value-based feature attribution method. Traditionally, Shapley values are defined as a conditional expectation, which is estimated via Monte Carlo sampling. This paper takes instead a Bayesian approach similar to that of [1], where the data is assumed to be generated from a weighted combination of Shapley values. As typical Bayesian procedures go, the Shapley values are now assigned priors, which have an associated posterior subject to inference. Therefore, feature attribution can be performed by inspecting the corresponding posterior.

The main distinction of the technique proposed in the paper is a more general formulation. The method of [1] was focusing on Gaussian processes, whereas the proceduce in the paper does not assume a specific type of model. For inference, the paper relies on black-box variational inference (VI). As typical in VI approaches, the model parameters, not just the Shapley values, can also be inferred through marginal likelihood maximization. The paper proposes tighter variational bound that should improve training of the model. Furthermore, to efficiently compute the subset marginal sufficient statistics, a masked embedding neural network is used.

1. Chau, S. L., Muandet, K., & Sejdinovic, D. (2023). Explaining the uncertain: Stochastic Shapley values for Gaussian process models. *Advances in Neural Information Processing Systems.*

**Additional Comments:**

On a technical note, the derivation of the classification model is unusual and could be simplified. In the paper, the probabilistic model for the classification model is defined as (I am using a more standard notation of defining the label as $y_i$ instead of $\mathbb{1}_i$ as in the paper):

$$
y_i \\sim \\mathrm{Bernoulli}({\\textstyle\\sum_{d}} \\varphi_{i,d}(x) + \\epsilon_i )
$$

Now, for classification, the "label noise" is already modeled by the Bernoulli likelihood. That is, there is nothing wrong with

$$
y_i \\sim \\mathrm{Bernoulli}({\\textstyle\\sum_{d}} \\varphi_{i,d}(x))
$$

In the context of classification, and more broadly Bayesian generalized linear models, the additional noise $ \\epsilon_i $ results in a "robust regression model." (See page 1172 of [1].) Unless this was intended, the robustification of the classification model does not seem to be necessary and could be ommited for clarity.
That is, the paper could instead use a unified representation of both regression and classification via

$$
y_i \\sim \\mathrm{likelihood}({\textstyle \sum_d}\\varphi_{i,d}(x)), \\theta) ,
$$

where $\\theta$ is an optional likelihood parameter. Regression (with continuous target variables) follows by setting $\\mathrm{likelihood}(y; \\hat{y}, \\sigma) = \\mathrm{Normal}(y; \\hat{y}, \sigma^2)$, while classification follows from $\\mathrm{likelihood}(y; \\hat{y}) = \\mathrm{Bernoulli}(y; \\hat{y})$. That is, no need for two separate models for regression and classification.

## References
1. Wang, C., & Blei, D. M. (2018). A general method for robust Bayesian modeling. *Bayesian Analysis*.

**Audience:**

Yes

**Audience Explanation:**

Yes, feature attribution is an important topic that covers an important susbet of the readership of TMLR.

**Claims And Evidence:**

No

**Claims Explanation:**

The written claims of the paper are a bit misaligned with what the paper is actually demonstrating in the experiments. First, the paper claims the following: The model is able to correctly retrieve the correct feature attribution given perfect model specification, variational inference (VI) provides an efficient and general way to infer Shapely values, and that the masked embedding neural network (MENN) is computationally efficient for Shapley feature attribution. Now, according to my reading of the paper, the main contribution is the use of the MENN for feature attribution. All the evaluations of probabilistic Shapley inference uses a MENN. In contrast, no evaluation of the "computational efficiency" of the VI procedure is conducted. Therefore, not all of the claims are properly evaluated.

Now, in my view, the use of VI is not particularly interesting or technically involved. The main contribution seems to be the use of the MENN, and VI is just an intermediate step to getting there. Therefore, I think the focus of the writing should focus on the use of the MENN, and explain why VI is necessary for this. This way, the current set of evaluations should be sufficient. However, most of the text will have to be rewritten and the title should better reflect the focus of the paper, which is MENN. (In fact, this is not the first paper to propose probabilistic Shalepy values, so it is overclaiming anayways. The previous work [1] already proposed a similar Bayesian formulation of Shapley values for example.)

1. Chau, S. L., Muandet, K., & Sejdinovic, D. (2023). Explaining the uncertain: Stochastic Shapley values for Gaussian process models. *Advances in Neural Information Processing Systems.*

**Requested Changes:**

## Major Comments
First, as mentioned above, I believe the paper should be re-organized to focus on the use of masked embedding neural network (MENN). In addition, the title should be changed to better reflect the new focus. (The paper is not the first to propose a probabilistic Shapley value framework, nor is that general.) Also, the paper, for some reason, seems to be avoiding the term "Bayes" in earlier section. I recommend using the term Bayes more liberally so that the positioning of the paper, which follows a Bayesian model of the Shapely values, can be more easily identified.

Furthermore the description of the probabilistic model in Section 3.1 needs to be improved.
* The paper should make it clear that previous works such as [7] already used a formulationa akin to Eq. (4). (I am not sure if [7] is the first paper to do this.) Also, the paper should focus on the differences with prior work in terms of modeling. For example, how is the prior on $\phi$ different from that in [7], except for the fact that they focused on Gaussian processes?
* The writing of Section 3.1 is a bit suboptimal. Just by reading the text, it is not obvious why Theorem 3.1 is relevant in this context. The Shapley kernel $P(S)$ is also not properly explained.

The mathematical expressions are confusing and lack of proper definitions in multiple places.
* Section 3 first stentence: Is $\mathcal{R}$ the usual Euclidean space or some arbitrary space? If it is the Euclidean space, why is it not $\mathbb{R}$?
* Eq. (1): In $f_d$, what is the meaning of the subscript $d$?
* Below Eq. (1): Where is $\mathbb{x}_s$ defined? What is this?
* Eq. (3): Where is $P(x | x_s)$ defined?
* $\mathcal{L}_{\mathrm{reg}}$ is a function of what?
* Eq. (11): Where is $\tilde{f}$ defined? Is it different from $f$?



Lastly, the paper does a poor job in terms of citing and discussing the relevant literature:
* The paper heavily involves variational inference, but does not cite a single paper on variational inference (VI). It is common to cite [1,2,3]. Furthermore, the paper refers to stochastic gradient-based VI as "stochastic variational inference" (4th paragraph in Section 1). This use of the term stochastic VI is confusing due to a highly infuencial paper of the same name [4] that proposes a very different type of algorithm. Instead, use something black-box VI [5] or stochastic gradient VI [6].
* Similarly, the paper does not cite anything for statements or references that necessitate a citation. For instance, in Section 1, "despite its desirable properties ... most existing Shapley-based feature attribution" which Shapely-based feature attribution? please cite some representative examples; "when model predictions are uncertain, deterministic Shapley values may give incomplete information on the confidence of ..." such claim needs a citation; "In addition, we are often not interested in identifying features with only high attribution, but also ..." this also needs a citation; "particular relevance in high-stakes domains such as healthcare or policymaking" also needs a citation; In Section 2 last paragraph, "Gaussian process" is used without citation.
* The relationship with previous works is not very well discussed. For example, the probabilistic model in Eq. (4) - (5) has already been considered in [7]. The main difference with the model in [7] must be discussed in more detail, especially for the prior on $\phi$.
* The formatting of the reference section is inconsistent. The venue of Adebayo *et al.* (2021) is not capitalized, Chau *et al.* (2023) was published in NeurIPS, "shap" in Broeck *et al.* (2022) should be capitalized, and so on. Please carefully review the reference section.

## Minor Comments
* Eq. (5): $\sigma(x)$ should be $\sigma(x)^2$.
* I recommend moving Section 2 to the back since it obstructs the flow from Section 1 to 3.
* Move the definition of the acronyms in Table 2 to the main text.

## References
1. Blei, D. M., Kucukelbir, A., & McAuliffe, J. D. (2017). Variational inference: A review for statisticians. Journal of the American statistical Association, 112(518), 859-877.
2. Jordan, M. I., Ghahramani, Z., Jaakkola, T. S., & Saul, L. K. (1999). An introduction to variational methods for graphical models. *Machine Learning*, 37(2), 183-233.
3. Zhang, C., Bütepage, J., Kjellström, H., & Mandt, S. (2018). Advances in variational inference. *IEEE Transactions on Pattern Analysis and Machine Intelligence*, 41(8), 2008-2026.
4. Hoffman, M. D., Blei, D. M., Wang, C., & Paisley, J. (2013). Stochastic variational inference. *Journal of Machine Learning Research*, 14(1), 1303-1347.
5. Ranganath, R., Gerrish, S., & Blei, D. (2014, April). Black box variational inference. In *Proceedings of the International Conference on Artificial Intelligence and Statistics* (pp. 814-822). PMLR.
6. Titsias, M., & Lázaro-Gredilla, M. (2014, June). Doubly stochastic variational Bayes for non-conjugate inference. In *International Conference on Machine Learning* (pp. 1971-1979). PMLR.
7. Chau, S. L., Muandet, K., & Sejdinovic, D. (2023). Explaining the uncertain: Stochastic Shapley values for Gaussian process models. *Advances in Neural Information Processing Systems.*

---

### Review · Reviewer_HHv4 · 2026-01-20

**Summary Of Contributions:**

The authors propose Probabilistic Shapley Inference (PSI), a framework that models feature attributions as latent random variables with input-conditional priors centered around Shapley values. PSI enables joint learning of a predictive model and its attribution distributions via a variational objective. A masked embedding neural network (MENN) architecture is introduced to handle variable-length inputs efficiently.

Framing Shapley values as means of latent random variables rather than point estimates addresses a genuine limitation in existing explainability methods, and the theoretical treatment is rigorous. The synthetic experiments in Section 4.1 demonstrate that PSI recovers true attribution distributions in controlled settings.

My main concerns relate to the validation of the core constraint, the missing quantitative evaluation of explanation and calibration quality, and limitations in scope and scalability analysis.

**On the core constraint $f_d = \phi_d$:** The authors acknowledge that PSI does not explicitly enforce this constraint but relies on the $D_{SHAP}$ regularization term (Section 5). The diagnostic in Remark 3.3 ($\mathbb{E}_x\{f_d(x)\} \rightarrow 0$) is stated as "necessary but not sufficient." I would like to see quantitative metrics (e.g., MSE between $f_d$ and true $\phi_d$) on synthetic datasets where ground truth is available, rather than only visual comparisons in Figure 3. On real datasets, correlation or rank agreement between learned attributions and post-hoc SHAP values would help validate this claim.

**On explanation quality evaluation:** The paper evaluates predictive performance (Table 2), architecture comparison via Jeffreys-Divergence (Figure 7), and localization on MNIST (Section 4.2), but there is no quantitative evaluation of the explanations themselves using standard metrics. Specifically:

- **Faithfulness metrics:** Infidelity and Faithfulness Correlation directly measure whether learned attributions $f_d$ align with actual model behavior. If PSI's claim that $f_d \approx \phi_d$ holds, INFD should be low and FCOR high.
- **Robustness metrics:** Average/Max Sensitivity measures stability under input perturbations. PSI's probabilistic framework should produce more robust attributions than point-estimate methods. This should be demonstrated.
- **Remove and Debiase (ROAD) metric:** Tests whether high-attribution features actually matter for predictions.

These metrics can be efficiently computed with small sample sizes (m=50-200) and would substantially strengthen the empirical claims. A comparison of PSI's $f_d$ against FastSHAP and KernelSHAP on these metrics is also missing to anchor performance.

**On uncertainty calibration:** The learned $\sigma_d(x)$ is presented as meaningful for decision-making (Section 4.2, Figure 6), but no calibration analysis validates this. I would like to see coverage statistics: what fraction of true $\phi_d$ values fall within 50%, 90%, and 95% credible intervals on synthetic data? Additionally, does high $\sigma_d$ correspond to genuinely unstable attributions as measured by sensitivity metrics? The uncertainty conflates multiple sources (data noise, model uncertainty, estimation variance) without disentangling them. Could you discuss how practitioners should interpret $\sigma_d$?

**On scope and scalability:** The limitation to binary classification significantly restricts applicability. The synthetic experiments are limited to D=3 features; I would like to see results for D=10, 20, 50 to demonstrate scalability of constraint satisfaction. No wall-clock timing comparisons are provided against FastSHAP or standard SHAP, making efficiency claims difficult to evaluate. The hyperparameter $\beta$ is searched over four orders of magnitude (0.001 to 10). How sensitive is performance to this choice?

**On baselines:** Table 2 compares predictive performance, but I am missing comparison to uncertainty-aware attribution methods. Chau et al. (2023), cited in the related work, propose a GP-based approach for Shapley uncertainty. A comparison of uncertainty calibration would be valuable. Bootstrap-based Shapley confidence intervals are also missing as a natural baseline for the uncertainty estimates.

**Audience:**

Yes

**Audience Explanation:**

Overall, the paper addresses an important problem with a probabilistic formulation and solid theoretical grounding.

**Broader Impact Concerns:**

No concerns.

**Claims And Evidence:**

No

**Claims Explanation:**

The theoretical claims in the submission are well-supported. The predictive performance claims are also convincingly backed by Table 2, which reports RMSE and PR-AUC.

However, my main concern is that the paper's central empirical claims, namely that learned attributions $f_d$ approximate true Shapley values and that $\sigma_d(x)$ provides meaningful uncertainty, lack direct quantitative validation. The synthetic experiments rely entirely on visual alignment between learned and ground-truth attributions; no quantitative metrics such as MSE or correlation coefficients are reported. The authors acknowledge in Section 5 that the diagnostic $\mathbb{E}_x\{f_d(x)\} \rightarrow 0$ is "necessary but not sufficient" for guaranteeing $f_d = \phi_d$, yet provide no stronger validation on real datasets where ground truth is unavailable.

The uncertainty calibration claim is only weakly supported. The paper presents illustrative examples on MNIST and ICU data, but does not report coverage statistics. I.e. what fraction of true $\phi_d$ values fall within learned credible intervals on synthetic data where ground truth is available? Without calibration analysis, it remains unclear whether $\sigma_d(x)$ provides reliable confidence estimates or conflates multiple uncertainty sources.

In summary, theoretical claims have strong support, predictive performance is convincingly demonstrated, but the core explainability claims, in particular that $f_d \approx \phi_d$ and that uncertainty is well-calibrated, are supported only by qualitative evidence and incomplete diagnostics rather than quantitative metrics.

**Requested Changes:**

**Critical for acceptance:**

1. **Add quantitative evaluation of explanation quality using established metrics.** The paper lacks any standard metrics for evaluating the learned attributions themselves. The authors should compute faithfulness metrics (Infidelity, Faithfulness Correlation) and robustness metrics (Sensitivity) on the learned $f_d(x)$ attributions and compare against baselines including FastSHAP and post-hoc KernelSHAP.

2. **Provide quantitative validation of the core constraint $f_d \approx \phi_d$.** On synthetic datasets where ground-truth Shapley values can be computed exactly, report MSE or MAE between $f_d(x)$ and $\phi_d(x)$ across the test set rather than relying solely on visual comparisons. On real datasets, report correlation or rank agreement between learned attributions and post-hoc SHAP values.

3. **Include uncertainty calibration analysis.** Report coverage statistics on synthetic data: what fraction of true $\phi_d$ values fall within 50%, 90%, and 95% credible intervals defined by $f_d(x) \pm z \cdot \sigma_d(x)$? Without this, claims that $\sigma_d(x)$ provides meaningful uncertainty for decision-making remain unsubstantiated.

**Would strengthen the work:**

4. **Add computational cost analysis.** Report wall-clock training and inference times compared to FastSHAP and standard neural network + post-hoc SHAP. The hyperparameter search over $\beta \in [0.001, 10]$ spanning four orders of magnitude suggests potential tuning overhead that should be characterized.

5. **Extend synthetic experiments to higher dimensions.** Current experiments are limited to D=3 features. Include D=10, 20, 50 to demonstrate scalability of constraint satisfaction and report quantitative metrics rather than only visualizations.

6. **Clarify uncertainty semantics.** Add discussion addressing what $\sigma_d(x)$ captures (data noise, model uncertainty, estimation variance) and whether practitioners should expect uncertainty to decrease with more training data.

---

### Review · Reviewer_Vx6T · 2026-03-09

**Summary Of Contributions:**

This work proposes a probabilistic framework that models and infers sufficient statistics of Shapley values. Based on this framework, it proposes a variational inference objective to simultaneously model Shapley values with a prediction task.
Due to the nature of Shapley values, this requires the removal of individual features during training, which is achieved through a proposed embedding-based architecture.
The work verifies the feature attribution performance of the approach in
empirical experiments on synthetic data, as well as its prediction performance on some real-world tabular data.

### Strengths

1. Modelling Shapely values and prediction task as a joint objective is an
   intuitive choice, as they are tightly related.
2. Providing Credible intervals to Shapley values is very meaningful for interpretability.
3. The work has a well written related work section.
4. The result has well-conducted empirical experiments comparing the proposed method to other interpretable baselines with respect to their prediction performance on tabular data.
5. The appendix is extensive and very helpful.

### Weaknesses

1. The work appears unpolished and hard to follow in places. This is mostly
   concerning Section 3, as well as most plots.
2. The majority of the empirical experiments is on synthetic data, and misses
   higher-dimensional applications such as vision and language.
3. There is no comparison to other feature attribution method on any
   established feature attribution benchmark.
4. There is no direct comparison of the proposed probabilistic Shapley
   inference on feed-foward neural versus the proposed masked embedding neural
   network on real-world datasets.

**Audience:**

Yes

**Audience Explanation:**

Providing credible intervals for Shapley values itself is of high interest for interpretability, which I think is nicely demonstrated in Section 4.2.
I also think that the idea of jointly modeling Shapely values and prediction task is very meaningful, into which this work provides some insight.

**Broader Impact Concerns:**

I do not see any ethical implications of this work that would require a broader
impact statement.

**Claims And Evidence:**

No

**Claims Explanation:**

There are no empirical results beyond synthetic data on the feature attribution performance for the proposed method, and it is not compared to any other feature attribution baseline.
Section 3 is hard to follow as it is very verbose.

**Requested Changes:**

*Critical:*
1. Section 3 is too verbose and hard to follow.
    1. This starts already with the definition of the variable-length subsets.
       An example can be found in Jethani et al. (2012) in Section 2, which
       define the same concepts without complicated notation.
    2. Theorem 3.1 only reiterates the properties of Shapley values, and can be
       safely moved to the appendix. Only Property 1 is used later.
2. The MENN architecture is very hard to understand. $R$ is not explained anywhere in the text, is it the same as $M_{1:K}$?
3. The empirical results in 4.4 miss a direct comparison of the feed-forward and the masked neural network architectures.
4. All quantitative results concerning the interpretability are done on synthetic data. For feature attribution, there are some established benchmarks that could be used to compare to other feature attribution methods. See for example Quantus by (Hedström et al., 2023).

*Stronger:*
1. The definitions of precision and recall described in Section 4.2 are too verbose and should be moved to the appendix.
2. Section 3.4.1 claims "nint(.) refers to the nearest integer function", yet in
  the appendix, it is explained that it randomly chooses to round up or down.
3. I do not think the title is a very good choice, as the "and" does not reflect that the proposed model does join Shapley and inference. Maybe something along the lines of "A variational framework to jointly learn Shapley values and prediction task.
4. Figure 4 should have the same color ranges for all plots for better comparability.
5. The legends in Figures 3 and 5 are obstructing the results and make them hard to read.
6. The axis labels of Figure 3 should be written out.
7. Figure 3 has different sizes of the ground truth plots and the estimated plots. Also, the estimated plots do not have y-axis labels.


References:

Hedström, Anna, et al. "Quantus: An explainable ai toolkit for responsible evaluation of neural network explanations and beyond." Journal of Machine Learning Research 24.34 (2023): 1-11.

---

### Note · Authors · 2026-04-16

I have read and agree with the venue's withdrawal policy on behalf of myself and my co-authors.